# Methionine consumption by cancer cells drives a progressive upregulation of PD-1 expression in CD4 T cells

Mahesh Pandit[1,7], Yun-Seo Kil[1,7], Jae-Hee Ahn[2,7], Ram Hari Pokhrel[1,7], Ye Gu[1], Sunil Mishra[1], Youngjoo Han[2], Yung-Taek Ouh [3], Ben Kang[4], Myeong Seon Jeong[5,6], Jong-Oh Kim [1], Joo-Won Nam [1], Hyun-Jeong Ko [2,8] ✉ & Jae-Hoon Chang [1,8] ✉

Programmed cell death protein 1 (PD-1), expressed on tumor-infiltrating T cells, is a T cell exhaustion marker. The mechanisms underlying PD-1 upregulation in CD4 T cells remain unknown. Here we develop nutrient-deprived media and a conditional knockout female mouse model to study the mechanism underlying PD-1 upregulation. Reduced methionine increases PD-1 expression on CD4 T cells. The genetic ablation of SLC43A2 in cancer cells restores methionine metabolism in CD4 T cells, increasing the intracellular levels of S-adenosylmethionine and yielding H3K79me2. Reduced H3K79me2 due to methionine deprivation downregulates AMPK, upregulates PD-1 expression and impairs antitumor immunity in CD4 T cells. Methionine supplementation restores H3K79 methylation and AMPK expression, lowering PD-1 levels. AMPK-deficient CD4 T cells exhibit increased endoplasmic reticulum stress and *Xbp1s* transcript levels. Our results demonstrate that AMPK is a methionine-dependent regulator of the epigenetic control of PD-1 expression in CD4 T cells, a metabolic checkpoint for CD4 T cell exhaustion.

Cancer cells import large amounts of nutrients from the tumor microenvironment (TME) to support biosynthetic demands associated with their survival and proliferation[1]. Therefore, the TME has always been a nutrition-competitive environment for cancer cells and immune cells. Metabolic changes and a nutrient-deficient environment in the TME promote tumor growth and impair antitumor immunity, exhausting T cells[2,3]. T cells are a focal point for activating the immune system against cancer, particularly owing to their capacity for antigen-directed cytotoxicity[4].

CD4 T cells show efficacious antitumor immunity by helping CD8 cytotoxic T lymphocytes (CTLs) or by direct antitumor activity[5,6].

Particularly, CD4 T cells and interferon (IFN)-γ control tumor growth without CD8 T cells[7]. Persistent exposure to cancer antigens without appropriate stimulation acts as a trigger for T cell exhaustion. Exhausted T cells show reduced effector function and different molecular-biological profiles, particularly increased surface expression of PD-1. Although CD8 T cell exhaustion has been well-established and considered a target of immunotherapy for cancer patients, relatively few studies have focused on the importance of PD-1 blockade for increased PD-1 expression and the functional recovery of CD4 T cells.

Amino acids are crucial for the normal function of cancer cells and T cells[8]. A nutrient-deprived TME is characterized by the limited

[1]College of Pharmacy, Yeungnam University, Gyeongsan-si, Gyeongsangbukdo 38541, Republic of Korea. [2]Department of Pharmacy, Kangwon National University, Chuncheon 24341, Republic of Korea. [3]Department of Obstetrics and Gynecology, School of medicine, Kangwon National University, Chuncheon 24289, Republic of Korea. [4]Department of Pediatrics, School of Medicine, Kyungpook National University, 68-Gukchaebosang-ro, Jung-gu, Daegu 41944, Republic of Korea. [5]Chuncheon Center, Korea Basic Science Institute (KBSI), Chuncheon 24341, Republic of Korea. [6]Department of Biochemistry, Kangwon National University, Chuncheon 24341, Republic of Korea. [7]These authors contributed equally: Mahesh Pandit, Yun-Seo Kil, Jae-Hee Ahn, Ram Hari Pokhrel. [8]These authors jointly supervised this work: Hyun-Jeong Ko, and Jae-Hoon Chang. ✉e-mail: hjko@kangwon.ac.kr; jchang@yu.ac.kr

utilization of amino acids such as glutamine, arginine, alanine, tryptophan, and methionine, which affects T cell survival, activation, differentiation, and acquisition of effector functions[8–10]. Leucine, an essential amino acid, is required for the immunological activation of CD4 T cells[11,12]. However, the degradation of tryptophan into kynurenine inhibits the activation of CD4 T cells and leads to their exhaustion[13,14]. Serine, glycine, and the one-carbon metabolism pathway contribute to the expansion of effector T cells[15]. Similarly, asparagine is crucial for the activation of CD8 as well as CD4 T cells[16]. In addition, targeting T cell metabolism can improve anti-cancer immunotherapy[17]. The roles of several amino acids on CD8 T cell regulation and effector function have been well-studied and reported; however, the effect of amino acids on the regulation of effector CD4 T cell functions are under-explored.

TME stifles T cell effector functions by upregulating immune suppressive proteins such as programmed cell death 1 (PD-1), cytotoxic T lymphocyte associated protein 4 (CTLA4), and T cell immunoglobulin domain and mucin domain 3 (TIM3) on T cells[18]. PD-1 is one of the hallmarks of T cell exhaustion, often associated with T cell dysfunction and unresponsiveness in cancer[19,20]. PD-1 signaling restricts initial immunological responses, maintains peripheral tolerance, and promotes tumor growth[19]. Blocking PD-1 signaling using anti-PD-1 therapy is effective anti-cancer immunotherapy[21].

Previous studies have reported ways of transcriptional modulation of PD-1 expression. Signaling through signal transducer and activator of transcriptions (STATs), nuclear factor of activated T cells 1 (NFATc1), c-Fos, and transforming growth factor (TGF)-β are known to induce PD-1 transcription[22,23]. GSK3β also induces PD-1 expression by suppressing T-bet expression[24]. On the contrary, T-bet inhibits PD-1 transcription by binding upstream of *Pdcd1*[25]. DNA methylation on the CpG site of *Pdcd1* is also associated with the direct transcription of

*Pdcd1*[26]. However, the metabolic mechanisms regulating PD-1 expression, such as factors regulating PD-1 in the TME, remain unknown. Here, we addressed how the high PD-1-expressing CD4 T cells are developed in the TME from a metabolic perspective.

In this work, we demonstrate methionine-dependent regulation of CD4 T cell exhaustion in the tumor microenvironment. Methionine consumption by cancer cells causes defects in the methionine cycle in CD4 T cells. Therefore, a sufficient supply of extracellular methionine in the TME is crucial for CD4 T cells to induce antitumoral immunity by limiting the expression of PD-1. Additionally, we identify epigenetic link between methionine metabolism and AMPK which downregulates PD-1 expression to enhance anti-tumoral immunity.

## Results

### Reduced level of methionine augments PD-1 expression in CD4 T cells

To determine the immune cells that exhibit higher PD-1 expression in the TME, we analyzed PD-1 expression in immune cells isolated from the draining lymph nodes of tumor-bearing or tumor-free mice. PD-1 expression was markedly upregulated in CD4, CD8, and regulatory T (Treg) cells, but not in B cells, from tumor-bearing mice, compared to those from tumor-free mice (Fig. 1a). Despite PD-1 expression being induced in T cells under inflammatory conditions, we observed higher PD-1 expression in tumor-infiltrating T cells than in T cells from tissues with severe inflammation (Supplementary Fig. 1a, b). Our findings suggest that the TME greatly upregulates PD-1 expression in T cells.

The deprivation of nutrients, accumulation of tumor metabolic waste, or excretion of exosomes from cancer cells in the TME may be associated with the upregulation of PD-1[27–29]. Therefore, we cultured immune cells in tumor-conditioned media (TM) obtained from B16F10 melanoma cells. We observed a significant upregulation of

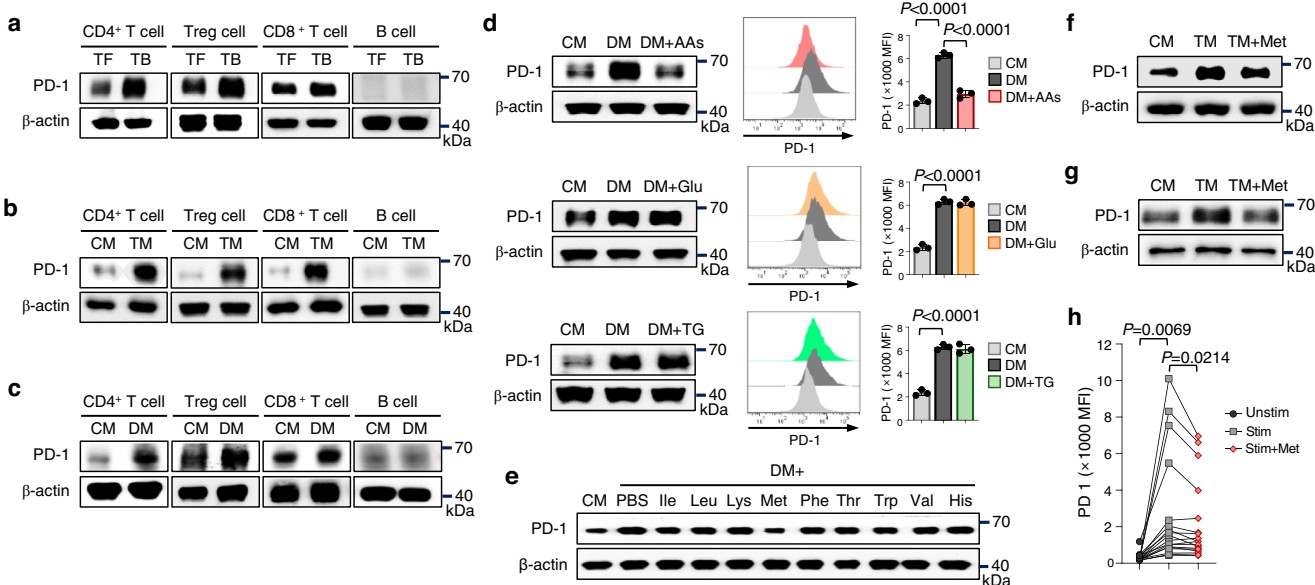

**Fig. 1 | Reduced methionine augments PD-1 expression in CD4 T cells.**
**a** Immunoblot analysis of PD-1 expression in CD4, Treg, CD8, and B cells isolated from the inguinal lymph node of tumor-free mice ($n = 6$) and dLN of tumor-bearing mice ($n = 6$). The experiment was performed three times. **b** Immunoblots showing PD-1 expression in activated immune cells were cultured in complete medium (CM) or tumor-conditioned medium (TM) for 72 h. **c** Immunoblots of PD-1 expression in cells cultured in CM or dialyzed medium (DM) for 72 h. The experiment was performed three times. **d** Effects of nutrient supplementation on PD-1 expression in DM-cultured CD4 T cells. Immunoblots for PD-1 and the mean fluorescence intensity (MFI) of PD-1 expression are represented in bar graphs ($n = 3$). The experiment was performed three times. **e** Immunoblots of PD-1 expression in CD4 T cells cultured in CM or DM supplemented with individual essential amino acids. The experiment was performed two times. **f, g** PD-1 expression on activated murine CD4 (**f**) and human CD4 T cells (**g**) cultured in CM or TM with or without methionine. The experiment was performed two times. **h** Effect of additional methionine treatment (200 µM) on PD-1 expression in activated human CD4 T cells ($n = 18$). Statistical analyses were performed using one-way analysis of variance (**d**). ANOVA followed by Holm-Sidak's multiple comparisons test (**h**). Data are presented as mean ± standard error of the mean. Source data are provided as a Source Data file.

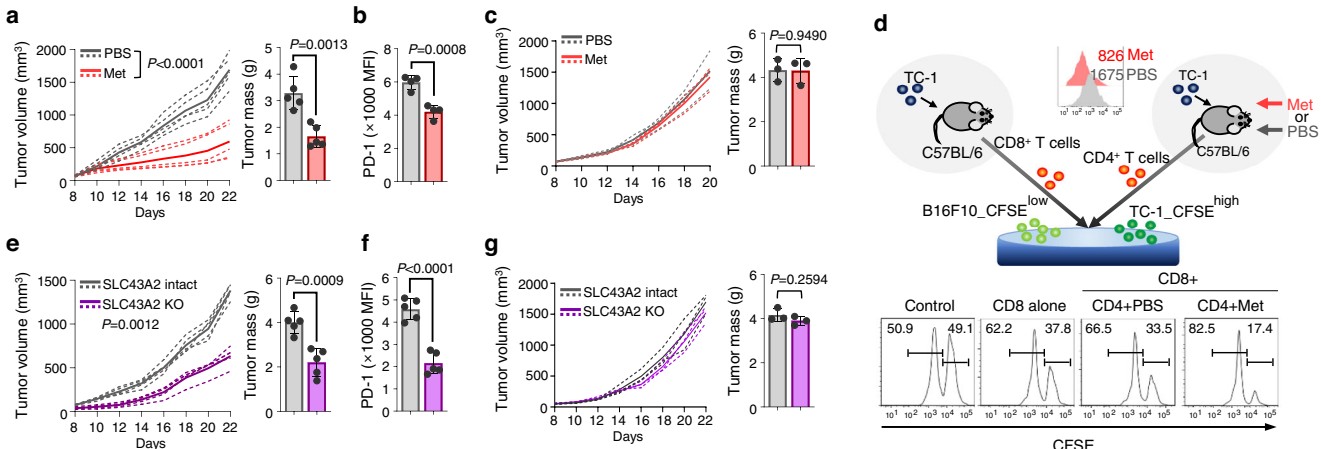

**Fig. 2 | Methionine supplementation enhances antitumor immunity.** Tumor volume and weight (n = 5 per group) (**a**) and PD-1 expression in tumor infiltrating CD4 T cells (n = 4 per group) (**b**) of mice transplanted with B16F10 melanoma cells and treated with methionine (40 mg/kg; intra-tumor) or control phosphate-buffered saline (PBS) every alternate day starting at day 8. **c** B16F10 tumor volume and tumor mass of Rag1⁻/⁻ mice who underwent PBS or methionine treatment every alternate day starting at day 8 (n = 3 per group). **d** Tumor-specific CD8 T cell-mediated cytotoxicity in vitro assay in the presence of CD4 T cells obtained from TC-1-bearing mice treated with PBS or methionine (n = 3 per group; the experiment was repeated twice). **e, f** Tumor volume and tumor mass (**e**) and PD-1 expression in tumor-infiltrating CD4 T cells analyzed using flow cytometry (**f**) in wild-type (WT) mice (n = 5 per group) injected with SLC43A2 KO or SLC43A2-intact B16F10 cells. **g** Tumor volume and tumor mass of SLC43A2-intact and SLC43A2 KO B16F10 cells were monitored in Rag1⁻/⁻ KO mice (n = 3 per group). Statistical analyses were performed using two-way analysis of variance (ANOVA) for tumor volume (**a, e**) and Two-tailed Student's t test (**a–g**). Data are presented as mean ± standard error of the mean. Source data are provided as a Source Data file.

PD-1 expression in CD4, CD8, and Treg cells cultured in TM following TCR and CD28 concomitant stimulation (Fig. 1b). Since rapidly proliferating tumor cells may outcompete T cells with high nutrient demands[30], we investigated whether the deprivation of nutrients was responsible for the upregulation of PD-1 in T cells. To do this, we cultured immune cells in complete medium (CM) or dialyzed medium (DM). When the immune cells were cultured in DM, the expression levels of PD-1 were similar to those of the TM cultures (Fig. 1c), indicating that reduced nutrition in the TME triggers PD-1 upregulation. Moreover, supplementation with essential amino acids in DM abrogated DM-induced PD-1 upregulation in CD4 T cells, whereas the addition of glucose and triglycerides did not (Fig. 1d). Supplementation with essential amino acids did not restrict the upregulation of PD-1 in CD8 T cells (Supplementary Fig. 1c–e), indicating that the upregulation of PD-1 expression in CD4, but not CD8 T cells, is associated with reduced levels of essential amino acids. Supplementation with methionine restrained PD-1 expression in activated murine CD4 T cells cultured in DM and TM (Fig. 1e, f) and activated human CD4 T cells cultured in TM (Fig. 1g). Similar results were obtained when activated human CD4 T cells were cultured in CM or CM supplemented with methionine (Fig. 1h).

### Methionine supplementation enhances antitumor immunity in a CD4 T cell-dependent manner

To assess the effects of methionine in vivo, B16F10 melanoma cells were transplanted into C57BL/6 mice. The intratumoral administration of methionine delayed tumor growth and reduced tumor mass (Fig. 2a and Supplementary Fig. 2a). The percentages of tumor-infiltrating CD4 and CD8 T cells as well as interferon (IFN)-γ and granzyme B (GZB) secretion were higher in the methionine-treated group than in the PBS-treated group (Supplementary Fig. 2b, c). Methionine treatment further reduced PD-1 expression in CD4 (Fig. 2b) but not in CD8 T cells (Supplementary Fig. 2d). B16F10-transplanted Rag1⁻/⁻ mice exhibited no alterations in tumor growth between the PBS- and methionine-treated groups (Fig. 2c), indicating that the antitumor effect of methionine was mediated by T cells. We performed an in vitro CTL killing assay using CD4 T cells with either high PD-1 (from PBS-treated tumor-bearing mice) or low PD-1 (from methionine-treated tumor-

bearing mice) expression levels (Fig. 2d). CFSEhigh-labeled TC-1 (right) and CFSElow-labeled B16F10 cells (left) were used as target and control cells, respectively (Fig. 2d). We observed that PD-1lowCD4 T cells enhanced the in vitro cytotoxic activity of tumor-infiltrating CD8 T cells isolated from TC-1 tumor against CFSEhigh-labeled TC-1 target cells compared to PD-1high CD4 T cells (Fig. 2d). In addition, when CD4 T cells were depleted in B16F10-transplanted mice, the increased antitumor effect and CD8 T cell activity caused by methionine treatment were reduced (Supplementary Fig. 2e–g), suggesting that methionine treatment enhances antitumor CD4 T cell immunity.

As the level of PD-1 expression on the T cells was reduced by methionine supplementation, a defect might have occurred in the methionine uptake efficiency of the T cells themselves or the extracellular methionine concentration might have decreased because of enhanced methionine use by the tumor. The levels of the main methionine transporters of CD4 T cells, including SLC7A5 and CD98[8], which form a heterodimeric structure comprising SLC7A5 and SLC3A2, were not markedly reduced in tumor-infiltrated CD4 T cells compared to those in CD4 T cells from tumor-free mice (Supplementary Fig. 2h, i). Therefore, we next assessed whether methionine uptake by cancer cells is responsible for the increased levels of PD-1 on CD4 T cells by lowering the extracellular level of methionine. SLC43A2 is a major transporter of methionine in several types of cancer cells[9]. We conducted a methionine uptake assay by adding methionine-d3 (200 μM) in the culture medium for 1 h. We found that intracellular methionine-d3 was lowered in SLC43A2 KO B16F10 cells than SLC43A2-intact B16F10 cells (Supplementary Fig. 2j). Compared to SLC43A2-intact B16F10 cells, SLC43A2 knockout (SLC43A2 KO) cells (Supplementary Fig. 2k) showed significantly delayed growth of transplanted tumors in wild-type (WT) mice (Fig. 2e). However, there was no difference in the in vitro proliferation of SLC43A2 KO and SLC43A2-intact B16F10 cells (Supplementary Fig. 2l). Furthermore, the infiltration of CD4 and CD8 T cells and IFN-γ and GZB secretions by these cells were markedly increased in SLC43A2 KO tumors compared to that in WT tumors (Supplementary Fig. 2m, n).

Interestingly, PD-1 expression in tumor-infiltrating CD4 T cells from SLC43A2 KO tumor-bearing mice was reduced (Fig. 2f and

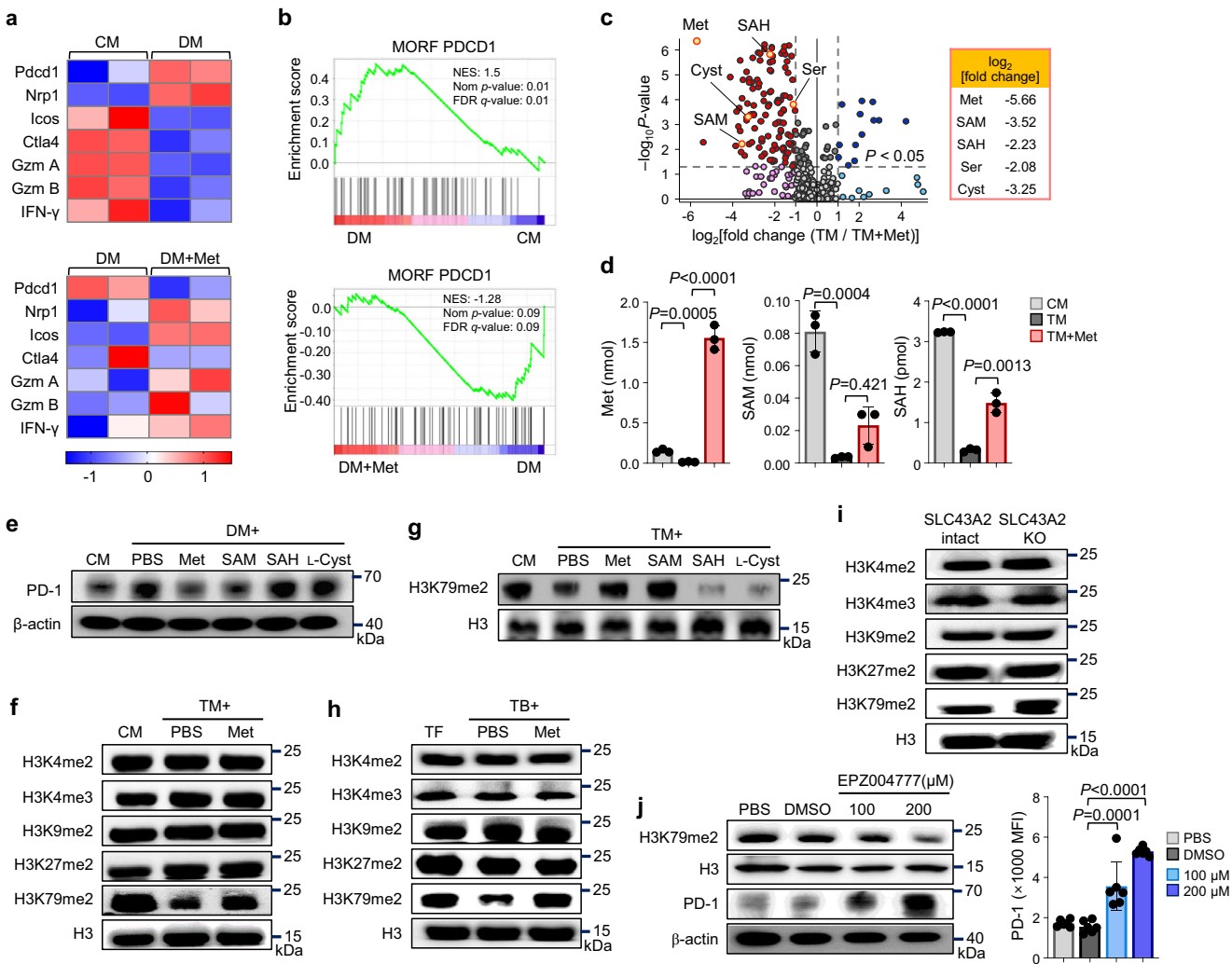

**Fig. 3 | Methionine restrains PD-1 via H3K79 methylation. a** Heat maps showing mRNA levels of antitumoral immunity-related proteins in CD4 T cells cultured in CM and DM alone or DM supplemented with methionine (*n* = 2 biological samples). **b** GSEA plot showing enriched PD-1 neighborhood genes in DM-cultured CD4 T cells. Enriched PD-1 neighborhood genes in CD4 T cells were downregulated by methionine supplementation (*n* = 2 biologically independent samples). **c** Volcano plot representing metabolic changes in CD4 T cells cultured in TM alone or TM supplemented with methionine (*n* = 3 independent samples per group). **d** Analysis of intracellular methionine, S-adenosyl methionine (SAM), and S-adenosyl homocysteine (SAH) concentrations via Liquid chromatography-mass spectrometry (LC-MS) in CD4 T cells cultured in CM, TM alone, and TM supplemented with methionine (*n* = 3 independent samples per group). **e** Immunoblots of PD-1 expression in CD4 T cells cultured for 72 h in CM, DM, or DM supplemented with methionine and its metabolites. The experiment was repeated two times. **f** Effects of TM and methionine treatments on the histone methylation profiles of cultured

CD4 T cells. The experiment was repeated three times. **g** Immunoblots for H3K79me2 in CD4 T cells cultured in TM with methionine and its metabolites. The experiment was repeated two times. **h** Effects of the TME and methionine treatment on histone methylation profiles. Immunoblots of CD4 T cells isolated from tumor-free (*n* = 6), PBS-treated tumor-bearing (*n* = 6), and methionine-treated tumor-bearing mice (*n* = 6). The experiment was repeated two times. **i** Histone methylation profiles of tumor-infiltrated CD4 T cells isolated from SLC43A2-intact and SLC43A2 KO tumor-bearing mice (*n* = 6 in each group). The experiment was repeated two times. **j** PD-1 and H3K79me2 immunoblots in CD4 T cells with DOT1L (EPZ004777) inhibitor treatment. The experiment was repeated two times. PD-1 MFI in CD4 T cells via FACS after DOT1L inhibitor treatment (*n* = 6 per group). Statistical analyses were performed using two-tailed Student's *t* test & TMM + CPM normalized method for correction (**b**), one-way analysis of variance (**d**, **j**) and Tukey HSD test (posthoc) after ANOVA (**c**). Data are presented as mean ± standard error of the mean. Source data are provided as a Source Data file.

Supplementary Fig. 2o), whereas that in CD8 T cells did not change significantly (Supplementary Fig. 2p). Growth retardation of SLC43A2 KO tumors was not observed in Rag1$^{-/-}$ mice (Fig. 2g). An analysis of The Cancer Genome Atlas (TCGA) database on RNA profiles obtained from melanoma, colon cancer and ovarian cancer revealed that *PDCD1* is positively correlated with *SLC43A2* (Supplementary Fig. 3a). The Kaplan–Meier survival curve suggested that patients from TCGA ovarian cancer dataset with lower levels of SLC43A2 transcripts showed a good prognosis for overall survival (Supplementary Fig. 3b). Human plasma methionine levels were negatively correlated with PD-1 expression in CD4 T cells ($r = -0.5878$, $p = 0.0131$) (Supplementary Fig. 3c). Thus, cancer cells outcompete CD4 T cells for methionine and

impair antitumor immunity in association with PD-1 upregulation in CD4 T cells.

## Low methionine level promptly decreases H3K79me2 level in CD4 T cells

An observation of the transcripts related to tumoral immunity revealed that *PDCD1* expression was upregulated in CD4 T cells cultured in DM (Fig. 3a). Similarly, gene set enrichment analysis (GSEA) showed enrichment of PD-1-related genes in T cells cultured in DM (Fig. 3b), whereas methionine supplementation largely rescued these transcript alterations (Fig. 3a, b). Moreover, the one-carbon metabolic process, methionine cycle, and pathways related to T cell immunity and

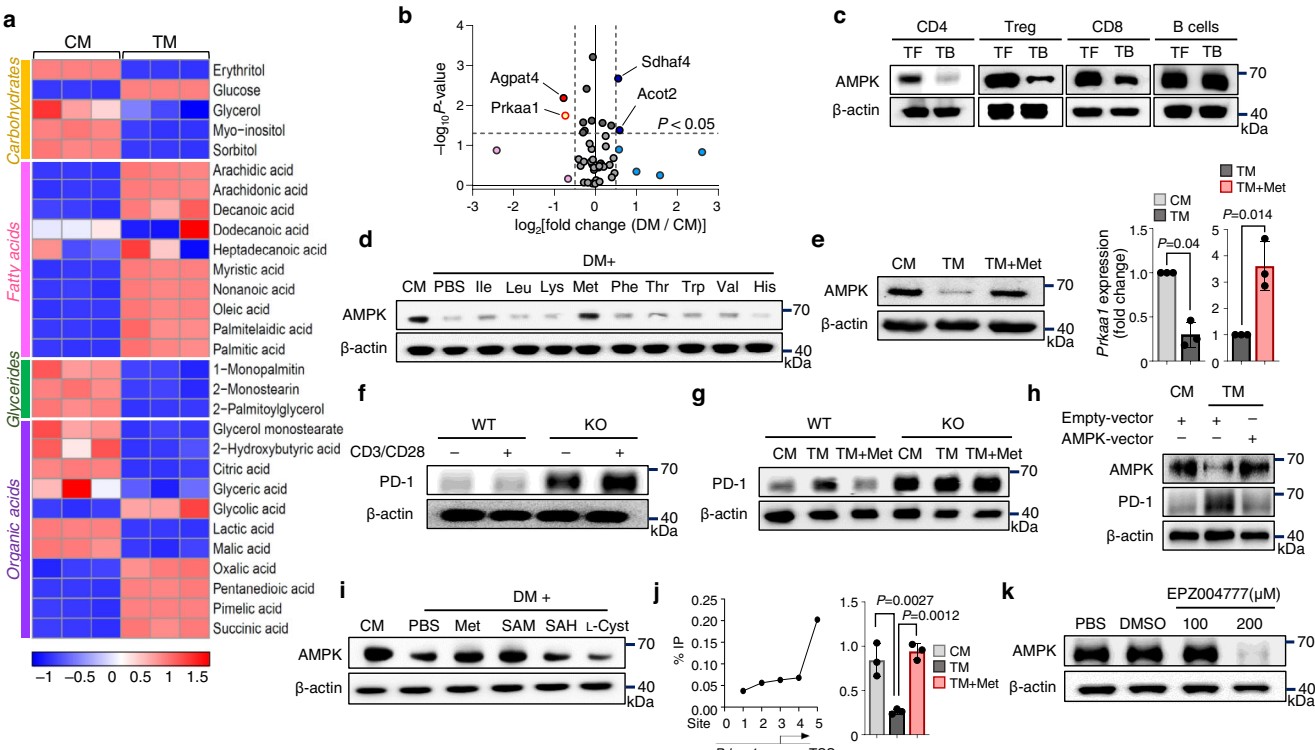

**Fig. 4 | Loss of H3K79me2 reduces AMPK expression. a** Metabolomic analysis of CD4 T cells cultured in CM and TM representing changes in carbohydrates, fatty acids, glycerides, and organic acids (n = 3 per group). **b** Volcano plot showing different metabolism-related genes in CD4 T cells cultured in CM and DM (n = 2 biologically independent samples). **c** Immunoblots showing AMPK expression in immune cells isolated from tumor-free (n = 6) and tumor-bearing mice (n = 6). The experiment was repeated three times. **d** Immunoblots showing AMPK expression in CD4 T cells cultured in CM, DM, or DM supplemented with individual amino acids. The experiment was repeated three times. **e** Expression of AMPK protein and *Prkaa1* in activated CD4 T cells cultured in CM, TM, and TM supplemented with methionine (n = 3 per group). **f** Immunoblots for PD-1 expression in unstimulated and stimulated CD4 T cells isolated from WT and AMPK KO mice. The experiment was repeated three times. **g** PD-1 expression in WT and AMPK KO CD4 T cells in CM and TM with or without methionine supplementation. The experiment was

repeated three times. **h** AMPK was overexpressed using a lentivirus vector in CD4 T cells cultured in CM and TM. AMPK overexpression and PD-1 expression were observed by western blotting. The experiment was repeated two times. **i** Immunoblot for AMPK in CD4 T cells cultured in CM, DM, and DM with methionine and its metabolites. The experiment was repeated two times. **j** Chromatin immunoprecipitation (ChIP) assay showing H3K79me2 occupancy in the *Prkaa1* promoter of CD4 T cells (left). ChIP assay showing H3K79me2 occupancy of the *Prkaa1* promoter in CD4 T cells cultured in CM, TM, and TM supplemented with methionine (right) (n = 3 per group). **k** Isolated CD4 T cells were cultured in the DOT1L inhibitor EPZ004777 for 12 h. Western blotting was performed for the detection of AMPK. The experiment was performed two times. Statistical analyses were performed using Tukey HSD test (posthoc) after ANOVA (**b**), two-tailed Student's *t* test (**e**) or one-way analysis of variance (**j**). Data are presented mean ± standard error of the mean. Source data are provided as a Source Data file.

cytokine production were downregulated in low nutrient conditions (Supplementary Fig. 4a–d).

We observed apparent metabolic changes in CD4 T cells cultured in TM compared with those cultured in CM, and these changes were partially rescued by the addition of methionine (Supplementary Fig. 4e, f). Extracellular methionine is transported into T cells through SLC7A5 and converted into S-(5-adenosyl)-L-methionine iodide (SAM), S-(5-adenosyl)-L-homocysteine (SAH), and L-cystathionine (L-Cyst)[31]. CD4 T cells cultured in TM showed a marked decrease in intracellular methionine, SAM, and SAH levels (Fig. 3c, d). Decreased levels of methionine and its metabolites were significantly restored when the CD4 T cells were supplemented with methionine (Fig. 3d). Furthermore, we observed that the increased PD-1 expression was restrained by the addition of methionine and SAM but not by SAH and cystathionine (Fig. 3e), suggesting a crucial role for methionine and SAM in PD-1 regulation in CD4 T cells.

SAM functions as a universal methyl donor in cells, especially for DNA and histone methylation[31,32]. We first assessed the level of DNA methyltransferase enzyme (DNMT) in T cells and found that it was reduced by TM and restored by methionine supplementation (Supplementary Fig. 5a). However, the treatment of activated CD4 T cells with decitabine, a DNMT inhibitor, did not alter PD-1 expression in CD4 T cells (Supplementary Fig. 5b, c). Next, we examined CD4 T cell

histone marks and found that culturing CD4 T cells in TM decreased demethylation at lysine 79 of histone H3 (H3K79me2) but did not alter the other histones assessed in this study (Fig. 3f). Methionine and SAM supplementation further restored H3K79 methylation in CD4 T cells (Fig. 3f, g). Similarly, CD4 T cells from B16F10-bearing mice exhibited decreased levels of H3K79me2, which were recovered by methionine supplementation (Fig. 3h). The level of H3K79me2 in tumor-infiltrated CD4 T cells was higher in SLC43A2 KO B16F10-bearing mice than in SLC43A2-intact B16F10-bearing mice (Fig. 3i). A previous study showed that restricting the extracellular methionine concentration to <10 μM depleted intracellular methionine levels in T cells, leading to a significant decrease in H3K4me3[31]. However, in our experiments, DM and TM contained >20 μM of methionine (Supplementary Fig. 5d) and did not affect H3K4me3 levels in TM (Fig. 3f). Therefore, reducing the extracellular levels of methionine to the extent that it can be found at physiological levels in the TME or culture supernatants (>20 μM) of cancer cells did not significantly alter H3K4me3 levels.

Next, we found that inhibiting H3K79 methylation with the inhibitor of DOT1L (EPZ004777), the only methyl transferase that can methylate H3K79[9,33], increased PD-1 expression in CD4 T cells (Fig. 3j). The consequences of histone methylation may be positive or negative in terms of transcriptional activity[34]. Surprisingly, an ENCODE study on public chromatin immunoprecipitation and sequencing (ChIP-seq)

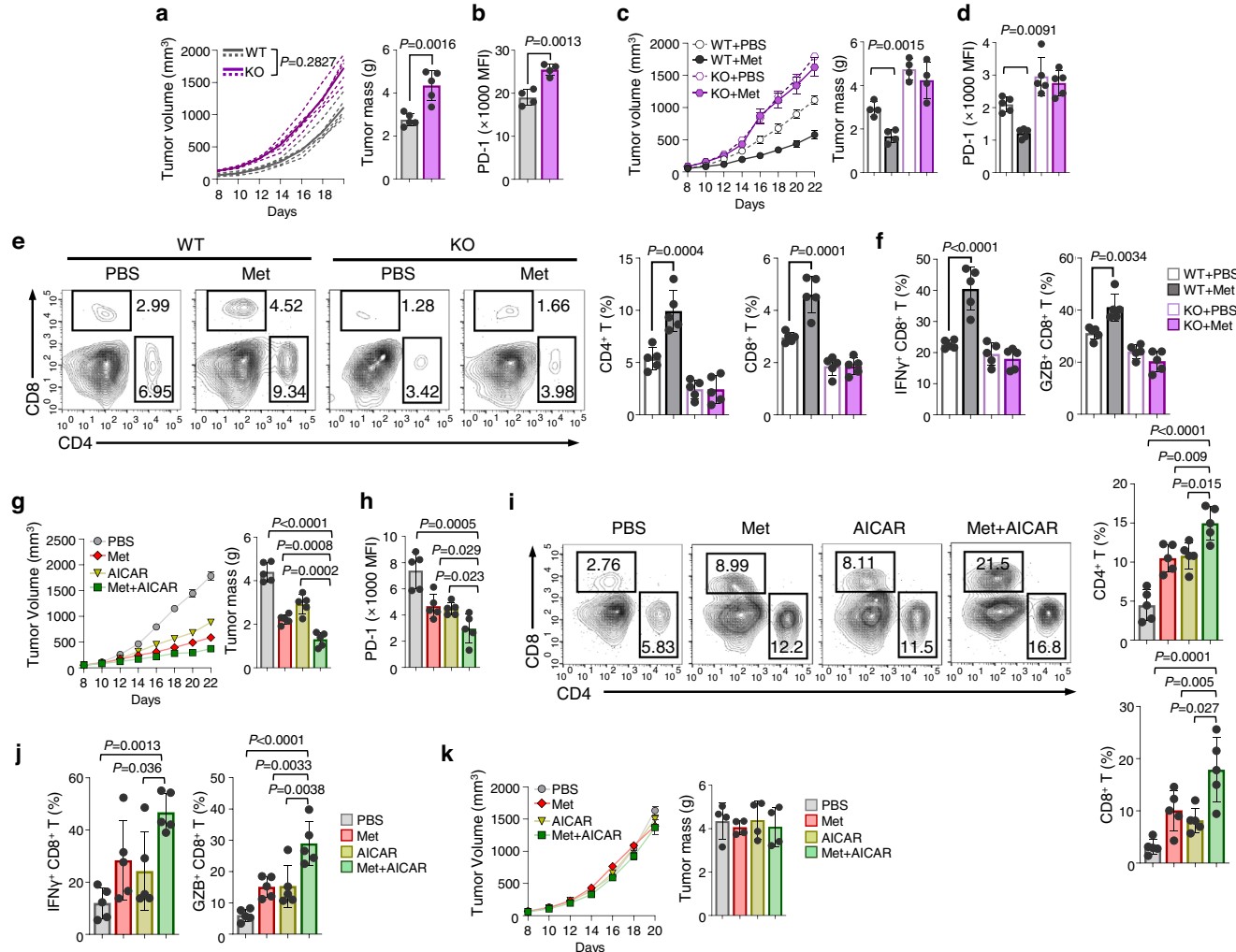

**Fig. 5 | AMPK activation in CD4 T cells enhances antitumor immunity.**
**a, b** B16F10 tumor volume and mass monitored in wild-type (WT) and AMPK KO mice (*n* = 5 per group) (**a**). Expression of PD-1 from CD4 TIL analyzed using FACS (*n* = 4 per group) (**b**). **c–f** B16F10 tumors were transplanted into WT or AMPK KO mice. Intratumoral injection of PBS or methionine (40 mg/kg) was administered every other day starting at day 8. Tumor volume was monitored daily, tumor weight was measured on day 22 (*n* = 4 per group) (**c**), and PD-1 expression in tumor infiltrating CD4 was analyzed using FACS (*n* = 5 per group) (**d**). Tumor-infiltrated CD4 and CD8 T cell percentages (*n* = 5 per group) (**e**). IFN-γ and GZB cytokine secretions by tumor-infiltrating T cells (*n* = 5 per group) (**f**). **g–j** B16F10 tumors transplanted in WT mice followed by treatment with methionine (40 mg/kg) or AICAR (500 mg/kg) alone or in combination every other day starting at day 8 (*n* = 5 per group). The tumor volume and weight were measured (**g**). PD-1 expression in tumor infiltrating

CD4 T cells was analyzed using FACS (**h**). Percentages of tumor-infiltrated CD4 and CD8 T cells (**i**). IFN-γ and GZB cytokine secretions by tumor-infiltrating T cells (**j**). **k** B16F10 tumor cells were injected into Rag1[−/−] mice, followed by treatment with methionine (40 mg/kg) or AICAR (500 mg/kg) alone or in combination every other day starting at day 8. Tumor volume and weight were monitored (*n* = 4 per group). Statistical analyses were performed using two-way analysis of variance (ANOVA) for tumor volume (**a, c, g, k**). (WT + PBS vs WT + Met *P* < 0.0001; KO + PBS vs KO + Met *P* = 0.4190 (**c**); PBS vs Met + AICAR *P* < 0.0001; Met vs Met + AICAR *P* < 0.0001; AICAR vs Met + AICAR *P* < 0.0001 (**g**)). two-tailed Student's *t* test (**a, b**) or one-way ANOVA multiple comparison tests were performed for bar graphs (**c–k**). Data are presented mean ± standard error of the mean. Source data are provided as a Source Data file.

data[35,36] revealed no binding of H3K79me2 on the PD-1 promoter in mouse and human cells (Supplementary Fig. 6a, b). These findings imply the indirect regulation of PD-1 expression by H3K79me2 and suggest that other transcripts may also regulate PD-1 expression.

### Reduced H3K79me2 impairs AMPK expression
Metabolites of the amino acid family were reduced in CD4 T cells cultured in DM and partially restored after adding methionine (Supplementary Fig. 4e). Moreover, the metabolites associated with the fatty acid family were higher, and those related to glycerides were lower in CD4 T cells cultured in DM than in those cultured in CM (Fig. 4a). These differences suggest that genes related to amino and fatty acid metabolic pathways may be involved in methionine-mediated PD-1 regulation. Thus, we next analyzed the genes related

to metabolic changes when cultured in less-nutritional conditions (DM) compared with when cultured in CM and found that the expression of *Agpat4* and *Prkaa1* was significantly reduced in CD4 T cells cultured in DM (Fig. 4b). *Prkaa1* encodes AMP-activated protein kinase subunit alpha 1 (AMPKα1), which is a key regulator of the glucose and lipid metabolic pathways[37]. Furthermore, AMPK can regulate PD-1 expression in Treg cells[38]. We, therefore, assessed the expression of AMPKα1 isolated from draining lymph nodes and observed reduced AMPKα1 expression in T cells but not in B cells from tumor-bearing mice (Fig. 4c). Furthermore, AMPK expression was lower in CD4 T cells cultured in DM or TM (Fig. 4d, e), and adding methionine rescued AMPKα1 expression in CD4 T cells (Fig. 4d, e). To assess the role of AMPK in the TME, we crossed mice carrying loxP-flanked *Prkaa1* alleles (AMPK[fl/fl]) with CD4[Cre] mice (hereafter referred to as "AMPK KO"). We

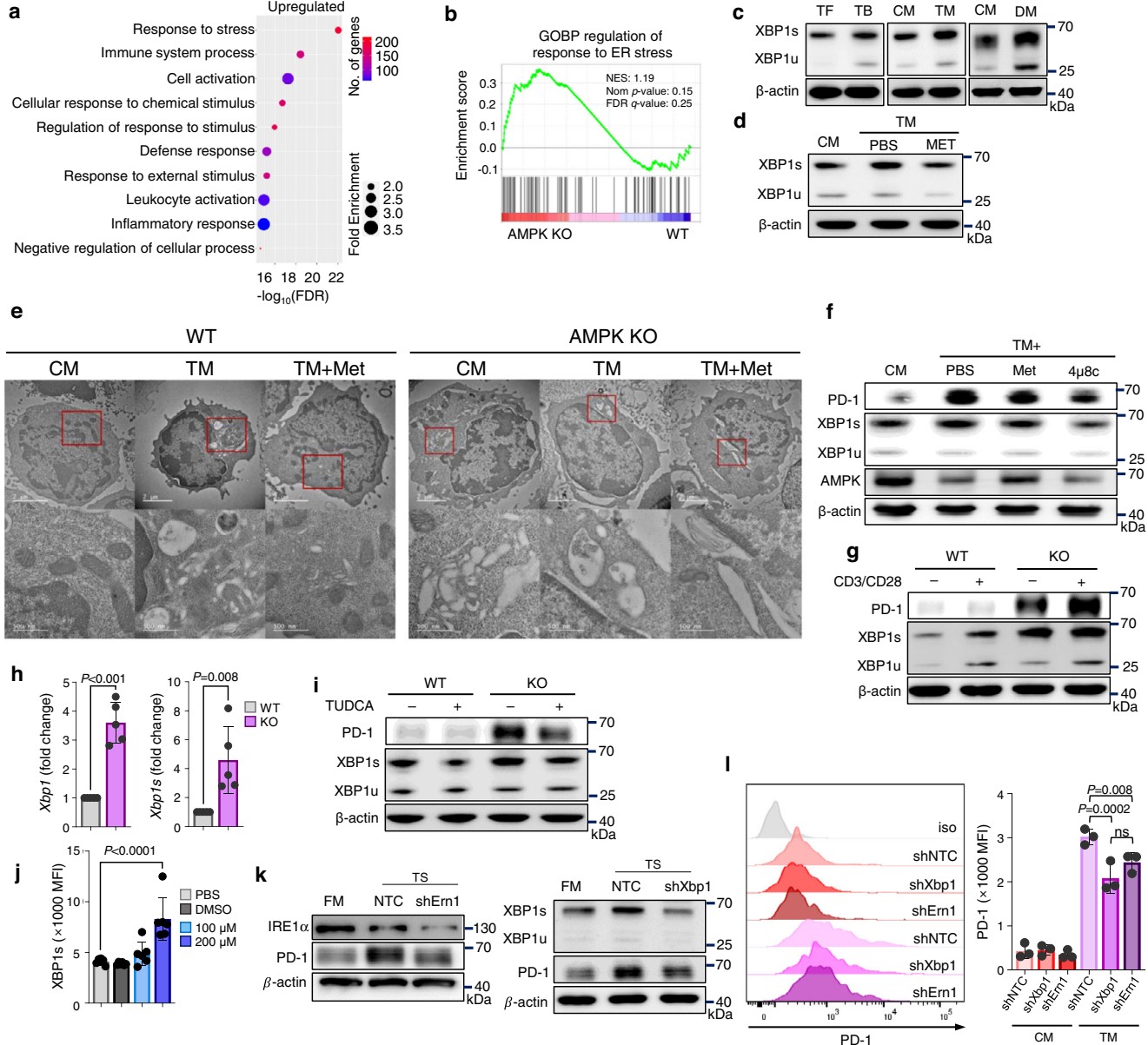

**Fig. 6 | AMPK regulates PD-1 via XBP1s. a, b** RNA-seq was performed on wild-type (WT) and AMPK KO CD4 T cells. Gene Ontology (GO) biological process analysis revealed the top 10 upregulated pathways in AMPK KO CD4 T cells. The x-axis is -log₁₀ (FDR); the different colors show the number of significantly regulated genes and the circles represent the fold-change in the enriched pathways. **b** GSEA plot of the response to ER stress-related genes. (**a**, **b**; n = 3 biological independent samples). **c** Immunoblots showing XBP1s and XBP1u expression in CD4 T cells from tumor-free and tumor-bearing mice (left panel) (n = 4 in each group) and cultured CD4 T cells in CM and TM (middle panel) or CM and DM (right panel). The experiment was repeated three times. **d** Immunoblot of XBP1s and XBP1u in CD4 T cells cultured in CM, TM, or TM supplemented with methionine. The experiment was repeated two times. **e** Representative TEM analysis showing the ER structure of WT or AMPK KO CD4 T cells in CM, TM, and TM supplemented with methionine. The size of scale bars is 2 μM (upper panels) and 500 nm (lower panels). **f** Immunoblots for PD-1, XBP1 and AMPK expression in CD4 T cells cultured in CM, TM, TM with methionine, and TM with an IRE1α inhibitor (4μ8c, 10 μM). The

experiment was repeated three times. **g** Immunoblots for PD-1, XBP1s and XBP1u in activated or non-activated WT and AMPK KO CD4 T cells. The experiment was repeated three times. **h** Fold-change in total *XBP1* and *XBP1s* transcripts in WT and AMPK KO CD4 T cells analyzed using RT-PCR (n = 5 per group). **i** Immunoblots for PD-1, XBP1s and XBP1u expression in WT and AMPK KO CD4 T cells with or without an ER stress inhibitor (TUDCA, 0.5 mM). The experiment was repeated three times. **j** Flow cytometry analysis of XBP1s in CD4 T cells with different doses of the DOT1L inhibitor (n = 6 per group). **k** Immunoblot analysis of PD-1 expression for *ERN1*(IRE1α) and *XBP1* knockdown CD4 T cells cultured in CM and TM. The experiment was repeated three times. **l** PD-1 expression in *XBP1* and *ERN1(IRE1α)* knockdown CD4 T cells determined using flow cytometry. (n = 3 per group). The experiment was repeated three times. Statistical analyses were performed using two-tailed Student's t test & TMM + CPM normalized method for correction (**b**), two-tailed Student's t test (**h**) or one-way analysis of variance (**j**, **l**). Data are presented as mean ± standard error of the mean. Source data are provided as a Source Data file.

observed higher levels of PD-1 expression in AMPK-deficient CD4 T cells than in WT CD4 T cells (Fig. 4f). However, methionine supplementation did not downregulate PD-1 expression in AMPK-deficient CD4 T cells (Fig. 4g), suggesting that methionine-mediated PD-1 regulation requires AMPK. To confirm this, we cultured CD4 T cells

overexpressing AMPKα1 in TM and found that PD-1 expression was maintained at a low level (Fig. 4h).

Next, we assessed the effects of methionine metabolites on AMPKα1 expression. Methionine and SAM restored AMPKα1 expression in CD4 T cells cultured in DM (Fig. 4i). Our chromatin

immunoprecipitation (ChIP) assay data and those from the public ChIP-seq study revealed high levels of H3K79me2 occupancy in the regulatory regions of the *Prkaa1* promoter (Fig. 4j and Supplementary Fig. 6c, d). Further, DOT1L inhibition led to a significant reduction in protein and mRNA expression of AMPK (Fig. 4k and Supplementary Fig. 6e). Thus, methionine induces the epigenetic methylation of histones in the *Prkaa1* promoter, and the expression of PD-1 is controlled by AMPK.

## AMPK activation in CD4 T cells enhances antitumor immunity

Compound C (CC) treatment prevented methionine-mediated reductions in tumor growth and PD-1 expression, resulting in enhanced tumoral immunity (Supplementary Fig. 7a–d). We inoculated B16F10 tumors into WT and AMPK KO mice to confirm this with a genetic approach. The AMPK KO mice exhibited a rapid progression in tumor growth and higher tumor weights (Fig. 5a). PD-1 expression on CD4 T cells was also higher in the AMPK KO mice than in the WT mice (Fig. 5b). Methionine did not affect tumor growth, PD-1 expression, or antitumoral immunity in the AMPK KO mice (Fig. 5c–f, Supplementary Fig. 7e, f).

Next, we administered 5-aminoimidazole-4-carboxamide ribonucleotide (AICAR), a pharmacological activator of AMPK, in combination with methionine, to B16F10- or MC38-bearing mice. The combined treatment significantly delayed tumor growth and enhanced antitumoral immunity by lowering PD-1 expression levels in tumor-infiltrating CD4 T cells (Fig. 5g–j, Supplementary Fig. 7f–h) but not in CD4 T cells isolated from lymph nodes of tumor-bearing mice (Supplementary fig. 7i). As we investigated the presence of other inhibitory markers, such as TIM-3 and LAG-3, there are no significant differences in the expression of TIM-3 and LAG-3 in CD4 T cells derived from lymph nodes of tumor-bearing mice (Supplementary fig. 7j), as well as in tumor-infiltrating CD4 T cells (Supplementary Fig. 7k) of tumor-bearing mice after treatment with methionine or AICAR supplementation or methionine and AICAR co-treatment. Further, the combined treatment of methionine and AICAR more suppressed tumor growth than methionine or AICAR treatment alone (Supplementary Fig. 7l), suggesting additive effect of methionine and AICAR even for SLC43A2-KO tumors. Comparable tumor growth in Rag1[−/−] mice after the methionine and AICAR treatment (Fig. 5k) was further observed, suggesting that the antitumoral effect induced by these compounds was mainly due to the effects of the T cells. Therefore, AMPK is a key molecule in methionine-mediated PD-1 regulation.

## AMPK regulates PD-1 via XBP1s

To investigate the mechanisms by which AMPK regulates PD-1, we performed RNA-seq on WT and AMPK-deficient CD4 T cells (Supplementary Fig. 8a). Gene Ontology (GO) analyses of differentially expressed genes revealed that the response to stress was most significantly enriched in AMPK-deficient CD4 T cells (Fig. 6a). Previous studies have suggested a role for endoplasmic reticulum (ER) stress in antitumoral immunity. They have further identified that the ER stress sensor XBP1 is involved in the expression of immune checkpoints, such as PD-1, in T cells[39]. In addition, GSEA showed enrichment in response to ER stress signature genes (Fig. 6b) in AMPKα1-deficient CD4 T cells compared to that in WT CD4 T cells. Most ER stress genes were further upregulated in AMPKα1-deficient CD4 T cells, and among them, *XBP1* was highly upregulated (Supplementary Fig. 8b). XBP1s, a spliced form of XBP1, binds directly to the promoter region of *PDCD1*[56]. Thus, XBP1 may be involved in methionine−AMPK axis-mediated PD-1 regulation. The level of total XBP1, that is, both XBP1s and unspliced form of XBP1 (XBP1u), was higher in tumor-infiltrating T cells and T cells cultured in TM or DM (Fig. 6c). Similar to PD-1, methionine supplementation restrained XBP1s and XBP1u expression in CD4 T cells cultured in TM (Fig. 6d). Transmission electron microscopy (TEM) analysis showed the expansion of the ER structure in CD4 T cells when they were cultured in TM, and methionine treatment restored the expansion of the ER structure in WT but not in AMPK KO samples (Fig. 6e). To assess the role of XBP1s in the regulation of PD-1 expression in CD4 T cells, we cultured CD4 T cells in the presence of the inositol-requiring enzyme 1α (IRE1α)-specific inhibitor 4μ8c. IRE1α is an ER stress sensor that transduces the unfolded protein signaling pathway[40]. Activating IRE1α mediates the conversion of the XBP1u mRNA into XBP1s, which acts as a transcription factor for the unfolded protein response in mammalian cells[41]. We found that the IRE1α inhibitor successfully reduced XBP1 splicing and inhibited PD-1 expression without affecting AMPKα1 expression (Fig. 6f). Thus, AMPK is an upstream regulator for XBP1s, and PD-1 is a downstream product. Furthermore, we performed a ChIP assay using CD4 T cells and found that in TM samples, XBP1s binds to the PD-1 promoter. TM+ Met samples showed reduced binding, suggesting that methionine prevents this binding (Supplementary Fig. 8c).

Next, we observed that the levels of XBP1s, XBP1u and PD-1 were increased in AMPK-deficient CD4 T cells (Fig. 6g). Similarly, real-time PCR revealed upregulated transcript levels of total XBP1 and XBP1s in AMPK-deficient CD4 T cells (Fig. 6h). When ER stress was inhibited using tauroursodeoxycholic acid sodium salt (TUDCA), the elevated PD-1 level in AMPK-deficient CD4 T cells was restored along with XBP1s reduction (Fig. 6i). TUDCA treatment to B16F10-bearing WT mice suppressed tumor growth (Supplementary Fig. 9a) and decreased PD-1 expression on CD4 T cells (Supplementary Fig. 9b). Furthermore, when we inhibited H3K79 methylation using EPZ00477 in CD4 T cells, XBP1s expression increased in a dose-dependent manner (Fig. 6j). Therefore, methionine-mediated H3K79me2 induces AMPK expression, which ultimately regulates PD-1 via XBP1s.

The knockdown of ERN1 (*IRE1α*) and *XBP1* reduced PD-1 in CD4 T cells cultured in TM compared to that in negative control short hairpin RNA (shRNA) (shNTC)-treated CD4 T cells which were cultured in TM. The reduction in PD-1 level was not significantly different between the shXBP1 and shERN1 knock-down groups (Fig. 6k, l). A recent study reported that increased extracellular cholesterol levels induce PD-1 via XBP1s in CD8 T cells[32]. To investigate these findings, we treated CD4 T cells with cholesterol. Interestingly, cholesterol treatment exclusively regulated PD-1 expression in CD8 but not in CD4 T cells (Supplementary Fig. 10a–c). The regulation of PD-1 by XBP1s in CD4 and CD8 T cells can therefore be caused by different mechanisms. Collectively, these data suggest that methionine regulates PD-1 via the AMPK−XBP1s axis in CD4 T cells.

## Discussion

CD4 T cells are essential for orchestrating anti-cancer immunity in the TME by promoting the killing effect of CD8 cytotoxic T cells[42]. Although many studies have reported the mechanisms underlying CD8 T cell exhaustion in the TME, few studies have assessed CD4 T cell exhaustion. Persistent exposure to cancer antigens induces CD4 T cell exhaustion by upregulating the expression of several co-inhibitory markers, including PD-1, CTLA-4, TIM-3, and Lag-3, and down-regulating the production of cytokines such as IFN-γ[18]. Similar to CD8 T cell exhaustion, the attenuated effector function of exhausted CD4 T cells may be recovered by treatment with an immune checkpoint blockade[43,44]. In this study, we found that the consumption of extracellular methionine by cancer cells induced T cell exhaustion by upregulating PD-1 expression in CD4 T cells. From a therapeutic perspective, methionine supplementation rescued the effector function of CD4 T cells from immunologically non-responsive conditions by downregulating PD-1 expression via increased AMPK expression.

Since activated T cells also require methionine for their function, cancer cells and activated T cells may compete to take up methionine. Increased SLC43A2 expression in various cancer cells can allow them to outcompete T cells for methionine uptake[9]. We found that reduced H3K79 methylation in tumor-infiltrated CD4 T cells reduced AMPK expression and consequently increased PD-1 expression. Reduced

extracellular methionine further downregulated AMPK expression in CD4 T cells, resulting in CD4 T cell exhaustion in the TME. Thus, we propose that supplementation with a sufficient amount of methionine in the TME can help T cells recover from an immunologically dormant state with increased AMPK expression.

Methionine is an essential amino acid indispensable for T cell activation and function. When activated, T cells require more methionine, which is metabolized to SAM, a universal methyl donor for DNA, RNA, and histone methylation. Gradual reduction of methionine level seems to preferentially affect specific histone methylation depending on methionine concentration. Restriction of extracellular methionine concentration to <10 μM depleted intracellular methionine levels in T cells, leading to a significant decrease in H3K4me3 level[31]. However, our findings revealed that lowering of extracellular methionine levels to physiological levels in the TME or culture supernatants of cancer cells did not alter H3K4me3 levels. In contrast, there was a significant decrease in the level of H3K79me2. Thus, H3K79me2 may be a highly sensitive epigenetic modulator for methionine than H3K4me3, recognizing subtle changes in the level of extracellular methionine. In addition, we found that PD-1 expression levels on CD4 T cells in the peripheral blood of ovarian cancer patients and healthy donors inversely correlate with serum methionine levels.

Upon activation by TCR signaling, CD4 T cells undergo metabolic reprogramming in that they show increased glycolysis and downregulate fatty acid oxidation. However, our metabolite analysis showed that T cells cultured in TM showed fatty acids and glucose accumulation and a decrease in lactate, suggesting the inhibition of both glycolysis and fatty acid oxidation. These findings were supported by RNA-seq data, which revealed a reduction in the expression of genes responsible for glycolysis and fatty acid oxidation in DM-cultured cells compared to those cultured in CM (Supplementary Fig. 4f). Altogether, our results suggest that decreased glucose metabolism and fatty acid oxidation in CD4 T cells cultured in DM are associated with reduced AMPK expression resulting from methionine depletion[45].

The accumulation of long-chain fatty acids, which is induced by the downregulation of AMPK[46] expression under methionine consumption by cancer cells, increases ER stress[47]. Methionine supplementation helped reduce ER stress by increasing AMPK expression, which decreased PD-1 expression in CD4 T cells. Furthermore, the alleviation of ER stress by TUDCA, 4μ8c, and shRNAs against IRE1α and XBP1 attenuated PD-1 expression. Here, the enhanced consumption of methionine by cancer cells facilitated the suppression of CD4 T cells by reducing AMPK expression. In contrast, the accumulation of cholesterol in the TME induces CD8 T cell exhaustion via high PD-1 expression[39]. Our findings suggest that methionine insufficiency-derived CD4 T cell exhaustion, which is mediated by ER stress-induced PD-1 expression, is different from CD8 T cell exhaustion, in which PD-1 expression is induced by the cholesterol–XBP1 axis[39].

T cell dormancy under methionine limitation may be induced by the following mechanisms. As shown in the results of this study, methionine limitation led to a decrease in the expression of AMPK by the reduction of H3K79 methylation. Next, ATP used for SAM production from methionine is reduced, resulting in a decrease in the AMP/ATP ratio and inhibiting the activity of AMPK. In addition, methionine is an amino acid encoded by the only start codon AUG, and hence, the initiation of protein synthesis is halted under methionine limitation. Finally, GCN2 activation induced by amino acid starvation, including methionine starvation, may inhibit protein translation by increasing the phosphorylation of eIF2α.

Upon activation, CD4 and CD8 T cells undergo proliferation and differentiate into effector cells. Effector CD8 T cells directly participate in the killing of target cells, whereas effector CD4 T cells elicited different helper T cell functions by producing different types of cytokines, suggesting that different regulatory mechanisms will be required for activated CD4 and CD8 T cells. Epigenetic regulation of T cell activity has also been proposed in metabolically distressed conditions, particularly with regard to methionine deficiency. Deprivation of methionine impaired CD8 T cell function and induced apoptosis, which can be reversed by adding methionine.

In summary, we reported methionine-dependent nutritional control of CD4 T cell exhaustion in the TME. Methionine consumption by cancer cells causes defects in the methionine cycle in CD4 T cells. Therefore, a sufficient supply of extracellular methionine in the TME is crucial for CD4 T cells to induce antitumoral immunity by limiting the expression of PD-1. Our data demonstrate an epigenetic link between methionine metabolism and AMPK, which can downregulate PD-1 expression to enhance antitumoral immunity, thereby delineating the role of methionine in CD4 T cell exhaustion in the TME.

## Methods

### Human specimen analysis and ethics statement
The specimens analyzed in this study were obtained from patients with and without ovarian cancer who had been diagnosed at Kangwon National University Hospital (KNUH, Chuncheon, Korea). The study protocol was approved by the Institutional Review Board (IRB) of KNUH (Chuncheon, Korea). Written informed consent was obtained from all patients. All methods were performed in accordance with the relevant guidelines and regulations of KNUH (IRB No. KNUH-2022-03-019). Peripheral blood mononuclear cells (PBMCs) were isolated[48]. Briefly, human peripheral blood was donated by patients and centrifuged in 3 mL of Histopaque-1077 (Sigma-Aldrich, St. Louis, MO, USA) at $400 \times g$ for 30 min. The upper supernatant plasma layer was collected, and the interphase containing PBMCs was washed thrice with phosphate-buffered saline (PBS). The PBMCs were stained with an anti-human specific antibody in FACS buffer (10 mM EDTA and 2% bovine serum-containing PBS) for 20 min. The following antibodies were used to stain the human PBMCs: FITC-conjugated anti-human CD4 antibody (1:100, Cat# 344604, RRID:AB_1937227) which was purchased from BioLegend (San Diego, CA, USA) and Alexa647-conjugated anti-human PD1 (1:100, Cat #560838, RRID:AB_2033988) antibody was purchased from BD Biosciences (Franklin Lakes, NJ, USA). To stimulate the T cells in the human PBMCs, 2 μL of Dynabead Human T-Activator CD3/CD28 were added to $2 \times 10^5$ PBMC culture in CM containing 10 ng/mL of human interleukin (IL)-2.

### Mice
All mice experiment was approved by Institutional Animal Care and Use Committee (IACUC) of Yeungnam University (Permit number 2014-018,2022-041 and 2023-009) and Kangwon National University (KW-220727-2). Mice were housed under specific pathogen-free conditions at the animal facility and handled according to the guidelines of the Institutional Animal Care and Use Committee (IACUC) of Yeungnam University and Kangwon National University. The animals were maintained in an animal facility at 20 °C to 22 °C with 40% to 60% relative humidity and a 12-hour/12-hour (light/dark) cycle for at least 7 days before the experiment. All mice were fed with normal standard diet containing 14% fat, 21% protein and 65% carbohydrate (5L79; Orient Bio, Inc., Seongnam, Korea). C57BL/6 J, Rag1$^{-/-}$, $CD4^{Cre}$, and $Prkaa1^{fl/fl}$ mice were purchased from Jackson Laboratory (Bar Harbor, ME, USA). Mice deficient in $Prkaa1$ in CD4 T cells (referred to as AMPK KO mice) and $Prkaa1^{fl/fl}CD4^{Cre-neg}$ (WT mice) were generated by breeding $Prkaa1^{fl/fl}$ with $CD4^{Cre}$ mice. Female mice (6–8 weeks old) were used in the experiments unless otherwise specified.

### Reagents and media
All essential amino acids, including L-methionine (Cat# M9625), SAM (Cat# A4377), SAH (Cat# A9384), L-cystathionine (L-cyst) (Cat# C7505), 2-amino-2norbornanecarboxylic acid (BCH) (Cat# A7902), Triglyceride Mix (TG) (Cat# 7810), Compound C (CC) (Cat# 171260), Tunicamycin (Cat# SML1287), glucose (Glu) (Cat# G8270), 5-Aza-2′-

deoxycytidine or decitabine (DAC) (Cat# A3656), and TUDCA (Cat# T0266) were purchased from Sigma-Aldrich. The DOT1L inhibitor EPZ004777 (Cat# 532282) and IRE1α inhibitor 4μ8c (Cat# 412512) were purchased from Merck Millipore (Burlington, MA, USA). B16F10 melanoma cells (CRL-6475™) and MC38 colon cancer cells were purchased from the American Type Culture Collection (ATCC) in 2006[49]. TC-1 cervical cancer cells[50] were kindly provided by Professor Tae Woo Kim (Korea University College of Medicine, Seoul, Korea). The cancer cells were maintained and amplified in vitro at 37 °C with 5% $CO_2$ in Dulbecco's modified Eagle's medium (DMEM; Cat# SH30243.01, Hyclone, Logan, UT, USA) supplemented with 10% fetal bovine serum (FBS; Cat# SH30919.03, Hyclone) and 1% penicillin/streptomycin (Cat# SV30010, Hyclone). For the TM, B16F10 tumor cells (CRL-6475TM, ATCC) were grown in RPMI 1640 medium (Cat# SH30255, Hyclone) containing 10% FBS and 1% penicillin/streptomycin". The media were extracted after attaining 80% confluence and used for the lymphocyte culture. For the DM, FBS was dialyzed to prepare dialyzed FBS (dFBS), in which the nutrients were reduced using dialysis tubing 3.5 kD (Cat# 086804 C, SpectrumLabs, San Francisco, CA, USA)[51]. DM was prepared by supplementation with 10% dFBS and 1% penicillin/streptomycin in RPMI 1640 medium without amino acids (Cat# R8999-04A, US Biological, Swampscott, MA, USA). CM consisting of RPMI 1640 medium (Cat# SH30027.01, Hyclone) with 10% FBS and 1% penicillin/streptomycin was used for the lymphocyte culture.

### Immune cell isolation and culture

CD4 and CD8 T cells, Treg cells, and B cells were isolated from the lymphoid organs of the mice using Miltenyi Biotec microbeads (Cat# 130117043 for CD4; Cat# 130116478 for CD8; and Cat# 130121301 for B cells; Miltenyi Biotec, Bergisch Gladbach, Germany) or the BD FACSJazz™ cell sorter (BD Biosciences, Franklin Lakes, NJ, USA). The isolated lymphocytes were stimulated with 2 μg/mL of anti-CD3 (Bio-Legend; Cat# 100359) and 2 μg/mL of anti-CD28 (Cat# BE0015-1, BioXcell, Lebanon, NH, USA) antibodies and cultured in vitro at 37 °C with 5% $CO_2$ in the indicated media depending on the experiments.

In brief, tumor-infiltrating lymphocytes (TILs) were isolated by mincing tumor tissues. Minced tumor tissues were digested using an enzyme mixture containing 0.5 mg/mL of Collagenase D (Cat# 11088866001, Roche Diagnostics, Basel, Switzerland) and 0.02 mg/mL of DNase I (Cat# 10104159001, Roche Diagnostics) in RPMI 1640 medium at 37 °C for 45 min with 200 rpm of continuous shaking. They were then passed through 100-μm nylon cell strainers (Cat# 352360, Falcon), and Percoll (Cat# 17-0891-01, GE Healthcare, Chicago, IL, USA) was then used to separate the TILs[52].

### Knockout of SLC43A2 in cancer cells

SLC43A2 knockout B16F10 cells were generated by deleting the *SLC43A2* locus with CRISPR/Cas9. To induce the ablation of *SLC43A2*, we cloned the lentiCRISPRv2 vector, which was a gift from Feng Zhang (Addgene plasmid #52961; http://n2t.net/addgene:52961; RRID: Addgene_52961[53]), with sgSlc43a2 oligos (sgSlc43a2 5′-TGTAAAGGC-GAAGGCCCGCA-3′). The sgSlc43a2-LentiCRISPRv2 plasmid was amplified in Stbl3™ (Cat# C737303, Invitrogen™, Waltham, MA, USA) on Luria–Bertani agar plates containing 50 μg/mL of ampicillin. Lentiviruses, which deliver sgSlc43a2 RNA, were produced in 293 T (CRL-3216™, ATCC) cells by co-transfecting 6 μg/mL of sgSlc43a2-LentiCRISPRv2, 4.5 μg/mL of psPAX2 (psPAX2 was a gift from Didier Trono; Addgene plasmid #12260; http://n2t.net/addgene:12260; RRID:Addgene_12260), and 1.5 μg/mL of pMD2.G (pMD2.G was a gift from Didier Trono; Addgene plasmid #12259; http://n2t.net/addgene:12259; RRID:Addgene_12259) using Lipofectamine™ 2000 Transfection Reagent (Cat# 11668027, Invitrogen™, Waltham, MA, USA) for 3 days. The virus-containing supernatant was collected, and B16F10 mouse melanomas were infected with the supernatant supplemented with 6 μg/mL of polybrene (Cat# TR-1003-G, Merck, Burlington, MA, USA).

Next, 24 h after the infection had been induced, the supernatant was replaced with fresh culture media containing 6 μg/mL of puromycin (Cat# A1113803, Gibco™, Waltham, MA, USA) to select B16F10 cells transduced with the virus. To identify the emergence of the indel sequence, the target sequence was amplified using the following primers: F: 5′-GCCCTTTCTCTACAGAGGTGAA-3′; R: 5′-ATTATCTTCCC TCTCCCCAAAC-3′. The amplified PCR product was cloned into a T/A cloning vector (Cat# RC001, Real-Biotech Corporation, Taipei, Taiwan). The isolated plasmid was sequenced using the M13 primer set (F: 5′-GTTTTCCCAGTCACGAC-3′; R: 5′-TCACACAGGAAACA-3′).

### Quantification of methionine from specimens with liquid chromatography-tandem mass spectrometry (LC-MS/MS)

The quantification of methionine in plasma donated from patients with ovarian cancer was conducted using LC-MS/MS (API3200, AB Sciex, Foster City, CA, USA)[54]. A mass spectrometry conjugated with an electrospray ionization (ESI) interface was used to produce positive ions. Optimized mass parameters of methionine and methionine-$d_3$ were determined by infusing each solution of the compounds at 1 ng/mL into the mass spectrometer in the positive ion mode at 10 μL/min. The chromatographic separation of methionine in the plasma was conducted with Gemini-NX 3μ C18 110 A New column 50 × 2.0 mm (656065-14) using an isocratic elution condition of 0.1% formic acid in water and methanol (50:50, v/v) at 0.3 mL/min and 40 °C using the Agilent 1200 series HPLC system (Agilent Technologies, Santa Clara, CA, USA). The analytical process was carried out using multiple reaction monitoring with an ion-spray voltage of +5500 V at 550 °C and selecting mass transitions of m/z 150.123 → 104.066 for methionine (declustering potential, 21 V; entrance potential, 6.5 V; collision energy, 13 V), 150.123 → 61.024 for methionine (21 V, 6.5 V, 31 V), 150.123 → 56.101 for methionine (21 V, 6.5 V, 25 V), and 153.058 → 63.964 for methionine-$d_3$ (56 V, 5 V, 31 V). The curtain and collision gases were 10 and 4 a.u., respectively. Unit resolution was used for both Q1 and Q3 quadrupoles. The data were analyzed using Analyst software (version 1.6). 50 μL of patient and normal human plasma containing 1, 5, 10, 50, 100, 500, 1000, 2000, 4000, and 8000 ng/mL of standard methionine was mixed with 150 μL of methanol containing 15 μg/mL of internal standard methionine-$d_3$ for 1 min. After centrifugation for 10 min at 12,000 × g and 4 °C, the supernatant was transferred into another LC vial and analyzed by injecting 5 μL of the supernatant into an LC-MS/MS operating system. The auto sampler was set to 10 °C during the analysis. For methionine uptake assay, 5×10$^6$ of WT (SLC43A2-intact) and SLC43A2 KO B16F10 cells were seeded in 100 mm cell culture dish. 200 μM of methionine-d3 (Cat#300616, Sigma) was added to culture medium. 1 h later, the cells were detached with 0.05% of trypsin-EDTA solution. Detached cell pellet was lysed in 50 μl of 80% methanol and centrifuged at 12,000 × g 10 mins. Supernatant was analyzed using LC-MS/MS with 20 ng/ml of carbamazepine (Cat #94496, Sigma) as an internal standard.

### Tumor inoculation and treatments

C57BL/6J, Rag1$^{-/-}$, AMPK KO, and WT mice were subcutaneously inoculated with 3 × 10$^5$ or 1 × 10$^6$ cancer cells per mouse and started the treatment when tumor volume reached around 50-150 mm³. Mice were euthanized by $CO_2$ inhalation when tumor volumes met humane endpoints described in the IACUC protocols or upon severe health deterioration. The maximum tumor diameter permitted under the relevant animal protocols is 20 mm, and this limit was not exceeded. Tumor volume was measured daily using a digital caliper in two dimensions (length and width) and calculated using the formula: $V = W^2 × L/2$, where $W$ and $L$ are the shortest and longest diameters, respectively, in mm. Methionine (40 mg/kg; intratumorally administered), AICAR (500 mg/kg; intraperitoneally injected) and TUDCA (150 mg/kg, intraperitoneally injected) treatments were administered

every day (TUDCA) and every other day (Methionine and AICAR), alone or in combination.

## Immunoblotting

Immunoblotting was performed[55]. Purified or cultured immune cells were lysed in radioimmunoprecipitation buffer (RC2002; Biosesang, Seongnam-si, Gyeonggi-do, South Korea) combined with 1% protease/phosphatase inhibitor (PPC1010; Sigma). The cell lysates were quantified using a bicinchoninic acid protein assay and resolved using sodium dodecyl sulfate-polyacrylamide gel electrophoresis. The resolved proteins were transferred onto polyvinylidene difluoride membranes (Immobilon®-P PVDF Membrane, IPVH00010, Merck) for 60 min and analyzed by immunoblots with the respective antibodies: anti-PD-1 (1:3000, D7D5W, Cat# 84651S; Cell Signaling Technology, Danvers, MA, USA, RRID:AB_2800041), anti-β-actin (1:3000, C4, Cat# sc-47778; Santa Cruz, RRID:AB_626632), anti-SLC3A2 (1:2000, Cat# MA5-29814; Invitrogen, RRID:AB_2785635), anti-SLC43A2 (1:2000, Cat# PA5-23571, Invitrogen, RRID:AB_2541071), anti-SLC7A5 (1:2000, Cat# PA5-50485, Invitrogen, RRID:AB_2635938), anti-DNMT1 (1:2000, 60B1220.1, Cat# ab13537; Abcam, RRID:AB_300438), anti-H3K4me2 (1:2000, Cat# ab32356; Abcam, RRID:AB_732924), anti-H3K4me3 (1:2000, Cat# ab8580; Abcam, RRID:AB_306649), anti-H3K9me2 (1:2000, Cat# ab1220; Abcam, RRID:AB_449854), anti-H3K27me2 (1:2000, Cat# ab24684; Abcam, RRID:AB_448222), anti-H3K79me2 (1:2000, Cat# ab177184; Abcam), anti-H3 (1:2000, Cat# ab1791; Abcam, RRID:AB_302613), anti-AMPKα1 (1:1000, Cat# 2532L; Cell Signaling Technology, RRID:AB_330331), and anti-XBP1s (1:1000, Cat# NBP1-77681SS; Novus Biological, RRID:AB_11060050) antibodies. The membranes were incubated with HRP-conjugated anti-mouse (1:4000, ADI-SAB-100-J, Enzo, Farmingdale, NY, USA, RRID:AB_11179634) or anti-rabbit IgG antibodies (1:4000, Cat# 7074, Cell Signaling Technology, RRID:AB_2099233) for 1 h at 20 °C. The blots were detected with a Fujifilm LAS-4000 mini (Fujifilm, Tokyo, Japan) using an enhanced chemiluminescence kit (Cat# 34580; Thermo Scientific).

## Flow cytometry analysis

For the flow cytometry analysis, cells were stained with fluorescence-conjugated monoclonal antibodies from BioLegend. Fluorescently labeled anti-CD3 (17A2), anti-CD8 (53-6.7), anti-CD4 (GK1.5, Cat# 100412 or RM4-5, Cat# 100528), anti-PD1 (RMP1-30, Cat# 109104 or 29F.1A12, Cat# 135224), LAG-3 (C9B7W, Cat# 125208), TIM-3 (B8.2C12, Cat# 134008), anti-IFN-γ (XMG1.2, Cat# 505808), anti-CD98 (RL388, Cat# 128208) antibodies from BioLegend (San Diego, CA, USA) anti-XBP1s (Q3-695, Cat# 562642) antibody from BD Biosciences (Franklin Lakes, NJ, USA) and anti-granzyme B (NGZB, Cat# 12-8898-82) antibody from eBioscience (San Diego, CA, USA) were used at 1:100 dilutions for the staining. For intracellular staining, the cells were stimulated for 4–6 h with ionomycin (750 ng/mL; Sigma-Aldrich) and phorbol 12-myristate 13-acetate protein 3 (50 ng/mL; Sigma-Aldrich) in the presence of Golgistop (10 μg/mL; BD Biosciences). After incubation, the cells were surface-stained, followed by fixation and permeabilization using a commercial buffer (BD Cytofix/Cytoperm;™ BD Biosciences) and intracellular protein staining. Stained samples were acquired using BD FACS Verse (BD Biosciences) and analyzed using FlowJo software, version 10.2 (BD Biosciences).

## In vitro CTL activity

To prepare the CTL target cells, TC-1 and B16F10 cells were labeled with high (5 μM) and low (0.5 μM) concentrations of carboxy-fluorescein diacetate succinimidyl ester (CFSE; C34554; Thermo Fisher Scientific, Waltham, MA, USA) for 15 min at 37 °C with gentle shaking in 5-min intervals. The CFSE^high-labeled TC-1 cells were then mixed with CFSE^low-labeled B16F10 cells at a ratio of 1:1, and $10^4$ cells of the mixed target cells were cultured in vitro with or without CD8 T cells ($5 \times 10^6$ cells) isolated from TC-1 tumor-bearing mice. CD4 T cells were also isolated from PBS- or methionine-treated TC-1 tumor-bearing mice and added to the respective groups. The cells were then cultured in vitro for 6 h at 37 °C before being analyzed using BD FACS Verse (BD Biosciences).

## Cell growth assay

Equal numbers of SLC43A2-intact and SLC43A2 KO B16F10 cells (10,000 cells/well) were cultured in a 96-well flat-bottom plate (Cat# 353072; Falcon) for 12 h. After the cells had adhered to the wells, the medium was discarded, and 100 μL of medium containing 0.5 mg/mL of 3-[4,5-dimethylthiazol-2-yl]-2,5-diphenyltetrazolium bromide (MTT) were added and incubated for 1 h at 37 °C. The medium was then discarded, and 100 μL of dimethyl sulfoxide was added to dissolve the formazan produced. Cell growth was determined by recording the absorbance at 570 nm using the SPECTROstar^Nano BMG LABTECH. The growth of the cell line was assessed after normalization and comparison with 12-h optical density values.

## RNA isolation and real-time PCR

The ReliPrep™ RNA Cell Miniprep System (Cat# Z6011, Promega Corporation, Madison, WI, USA) kit was used to isolate total RNA from purified CD4 T cells. cDNA was synthesized using the Goscript Reverse Transcription system (Cat# A5001, Promega Corporation). The mRNA expression level of each gene was measured using real-time PCR using the QuantiTect SYBR Green PCR kit (Qiagen, Hilden, Germany). Relative gene expression was determined using the $C_t$ value[56].

$$2^{-\Delta\Delta C(t)} = [(C_t \text{ of the gene of interest} - C_t \text{ of internal control})\text{Sample A} \\ - (C_t \text{ of the gene of interest} - C_t \text{ of internal control})\text{Sample B}] \quad (1)$$

The list of primers and their sequences are presented in Supplementary Table 1.

## ChIP assay

CD4 T cells ($1–2 \times 10^7$) were fixed with 1% paraformaldehyde for 10 min and quenched with 125 mM glycine. The fixed cells were washed with ice-cold PBS three times and resuspended in ChIP assay buffer (50 mM HEPES, 140 mM NaCl, 1 mM EDTA, 1% Triton X-100, 0.1% SDS, and 0.1% sodium deoxycholate). The cells were sonicated three times at 30% amp for 10 s (Fisherbrand™ Model 120 Sonic Dismembrator, Fisher Scientific, Waltham, MA, USA). Immunoclearing was performed using a protein A bead (PRT-7100, Repligen, Waltham, MA, USA) slurry for 2 h at 4 °C. After removing the slurry by centrifugation at $2000 \times g$, the supernatant was reacted with non-specific IgG (Cat# ab171870, Abcam) and anti-mouse H3K79me2 antibodies (Cat# 39143, ActiveMotif, Carlsbad, CA, USA, RRID:AB_2561018) or anti-mouse XBP1s antibody (Cat# 82914, Cell signaling Technology) overnight at 4 °C. The next day, 50 μL of protein A-coated slurry was added for conjugation with the antibodies. After centrifugation, the beads were washed with low-salt buffer (0.1% SDS, 1% Triton X-100, 2 mM EDTA, 20 mM Tris-HCl, 150 mM NaCl), high-salt (0.1% SDS, 1% Triton X-100, 2 mM DETA, 20 mM Tris-HCl, 500 mM NaCl), LiCl (0.25 M LiCl, 1% NP-40, 1% sodium deoxycholate, 1 mM EDTA, 10 mM Tris-HCl) buffer, and TE (10 mM Tris, 1 mM EDTA) buffer. DNA was eluted using elution buffer (1% SDS, 100 mM NaHCO_3). Reverse cross-linking was conducted overnight at 65 °C. DNA was purified using a PCR purification kit (YDS100, Unk Biotech Corporation, Taipei, Taiwan). The following primers were used to quantify the precipitated DNA: Prkaa1-ChiP-F; 5′-GATGTGCGGTGTCCCATAGT-3′, Prkaa1-ChiP-R; 5′-CATCTGGAAGCCA AGGGTGT-3′, Xbp1s-ChiP-F; 5′-AATCCCAGAGAGACAAGCAGGAG-3′, Xbp1s-ChiP-R; 5′-TTCCCCTGAGAAAAACCTAACA-3′. Real-time PCR was performed using the GoTaq® qPCR system (A6001, Promega, Madison, WI, USA). The results were calculated using the % IP method. % IP = $100 \times 2^{(Ct \text{ of specific Ab} - Ct \text{ of input})}$.

## Knockdown and overexpression of target proteins

To generate shRNAs expressing lentiviruses for silencing mouse *Xbp1* and *Ern1*, we cloned the sequences 5′-CCATTAATGAACTCATTCGTT-3′ (for *Xbp1*) and 5′-GCTCGTGAATTGATAGAGAAA-3′ (for *Ern1*) into the pLKO-Thy1.1 vector (gifted by Christophe Benoist and Diane Mathis; Addgene plasmid #14749; http://n2t.net/addgene:14749; RRID: Addgene_14749[57]). To overexpress AMPKα1 in CD4 T cells, we purchased the pLenti.CMV/TO_PRKAA1 vector from Addgene (pLenti.CMV/TO_PRKAA1 was a gift from Reuben Shaw; Addgene plasmid #74446; http://n2t.net/addgene:74446; RRID: Addgene_74446[58]). The obtained lentivirus plasmid was transfected into 293 T cells (CRL-3216™, ATCC), as described above.

Isolated CD4 T cells from naïve mice were expanded for 2 days with plate-bound anti-CD3 and anti-CD28 with 10 ng/mL of IL-2 (Cat# 212-12, Peprotech, Rocky Hill, NJ, USA). The CD4 T cells were infected with concentrated viruses and 10 μg/mL of polybrene (Cat# TR-1003-G, Merck) by centrifuging at 1500 × *g* for 90 min. Approximately 24–48 h after the lentiviral infections, the proteins were extracted using RIPA buffer.

## Metabolomics

Samples for metabolomics analysis were prepared with slight modifications[59]. Briefly, CD4 T cells were cultured in CM, TM alone, and TM supplemented with 100 μM methionine together with anti-CD3 (2 μg/mL) and anti-CD28 (2 μg/mL) stimulation for 72 h at 37 °C with 5% CO$_2$. After incubation, the supernatant was discarded, and the cell pellet was dissolved in 750 μL of 80% cold methanol (maintained at −80 °C for 30 min). The mixture was vortexed vigorously for 20–30 s and maintained at -80 °C for 30 min. Samples were centrifuged at 12000 × *g*, and the supernatant was collected in microcentrifuge tubes. Again, 250 μL of cold methanol was added, and the mixture was vortexed and maintained at −80 °C for 10 min, followed by high-speed centrifugation. The supernatants were collected in new tubes, quantified, and normalized by protein concentration. The samples were dried in a SpeedVac vacuum centrifuge (EYELA, Tokyo, Japan) without heat to remove methanol and kept at -20 °C.

For the liquid chromatography-mass spectrometry (LC-MS) analysis, the dried sample pellets were resuspended in 20 μL of LC-MS grade water, ultra-sonicated for 10 min, and centrifuged at 12000 × *g* for 10 min. Supernatants were transferred into an LC-MS vial for sample injection. LC high-resolution (HR) ESIMS/MS data were acquired on a Thermo Scientific Q Exactive hybrid Quadrupole-Orbitrap mass spectrometer coupled to a Thermo Scientific Vanquish UHPLC System (Waltham, MA, USA) at the Core Research Support Center for Natural Products and Medical Materials (CRCNM), equipped with a Waters XBridge Amide 3.5 μm column (4.6 × 100 mm, Milford, MA, USA). The gradient mobile phase system was composed of 0.1% formic acid in water (A) and 0.1% formic acid in acetonitrile (B): 0 − 18 min (85 − 45% B), 18 − 19 min (45 − 30% B), 19 − 21 min (30% B isocratic for washing), 21 − 22 min (30 − 85% B), and 22 − 30 min (85% B isocratic for re-equilibrating)[60]. The flow rate was 0.5 mL/min. The injection volume was 5 μL for each sample. The column temperature was maintained at 30 °C. The positive-ionization mode was used for MS detection with an *m/z* range of 100–1000. MS/MS fragmentation was achieved for the three most intense parent ions by collision-induced dissociation at 30% normalized collision energy. The obtained LC-MS/MS data were processed in Compound Discoverer (Thermo Scientific) for metabolite annotation using databases including mzCloud and Metabolika. Volcano plot comparisons were performed between TM alone and TM supplemented with methionine for the annotated metabolites. The data matrix was exported from Compound Discoverer, and a heat map for selected metabolites was generated using the "pheatmap" package in R software (version 4.1.0). The LC-MS analysis was also performed to measure intracellular methionine, SAM, and SAH

in the indicated media. Calibration samples of methionine, SAM, and SAH were prepared with LC-MS grade water by serial dilutions of stock solutions to give 6−10 calibration points. Injection volume was set as 1 (for methionine detection) or 5 μL (for SAM and SAH detection). The acquired LC-HRESIMS data were processed in FreeStyle 1.6 (Thermo Scientific) software to generate extracted ion chromatograms at *m/z* 150.0583, 399.1447, and 385.1289 for the detection of methionine, SAM, and SAH, respectively, with 5 ppm of a mass tolerance and to obtain peak area values (auto-peak detection option was applied).

For gas chromatography-mass spectrometry (GC-MS), silyl derivatization was conducted prior to analysis to improve the volatility and thermal stability of analytes. Samples were analyzed using an Agilent 7890 A GC (Agilent Technologies) coupled to a LECO Pegasus III time-of-flight (TOF) MS (St. Joseph, MI, USA) with a Restek Rtx-5MS (30 m × 0.25 mm I.D. × 0.25 μm film thickness, Bellefonte, PA, USA). The injection volume was 1 μL. The TOF-MS data were recorded in the *m/z* range of 50–1000 with a gas (helium) flow rate of 1.5 mL/min and a split ratio of 10. Relative quantities of detected metabolites were visualized as a heat map in R. The metabolomics datasets are provided as supplementary data 1 and 2.

## RNA-seq and bioinformatics analyses

CD4 T cells sorted from C57BL/6 J mice were cultured in CM, DM, and DM supplemented with methionine (DM + Met) for 72 h at 37 °C, along with anti-CD3 (2 μg/mL) and anti-CD28 (2 μg/mL). CD4 T cells isolated from WT and AMPK KO mice were first rested for 1 h in RPMI 1640 medium and stimulated with anti-CD3 (2 μg/mL) and anti-CD28 (2 μg/mL) for 6 h at 37 °C. After incubation, the cells were harvested, and 750 μL of TRIzol™ (15596026, Invitrogen) was added. The prepared samples were transferred to Ebiogen Inc. (Seoul, South Korea) and maintained at -80 °C for further experiments and analyses.

## Utilization of publicly available data from TCGA

The results are shown in whole or part based on data generated by TCGA Research Network: https://www.cancer.gov/tcga. The RNA expression datasets were obtained from https://www.cbioportal.org/[61,62]. A correlation analysis between the expression of *SLC43A2* and *PDCD1* was performed.

## ENCODE study

Public ChIP-seq data were obtained from the ENCODE project. The ENCODE integrative analysis [63] and ENCODE portal[64] were used in this study. We acknowledge the ENCODE Consortium and ENCODE production laboratory(s) for generating the dataset(s). We downloaded the call sets from the ENCODE portal[65] (https://www.encodeproject.org/) using the following identifiers: ENCFF076DCC, ENCFF219JIE, ENCFF225RZA, ENCFF063WGZ, and ENCFF042INO.

## TEM analysis

CD4 T cells were dehydrated in a graded ethanol series, treated with a graded propylene oxide series, and embedded in Epon (TED Pella Inc., Redding, CA, USA). The embedded cells were sectioned into ultrathin slices (80 nm) and placed on copper grids. The sliced cells were stained with uranyl acetate and lead citrate. The stained sections were observed using TEM (JEM-2100F; Jeol, Tokyo, Japan) at 200 kV. The analysis was performed at the Korea Basic Science Institute (KBSI) Chuncheon Center.

## Statistical analysis

For experimental data, normal distribution was tested, and statistical analyses were performed using GraphPad Prism software (version 8, GraphPad Software Inc.). Data are presented as the mean ± standard error of the mean. Comparisons between two groups were assessed using two-tailed Student's *t* tests. Multiple comparisons were assessed

**Article**

using a one-way analysis of variance, including Tukey's multiple comparisons test. $P$ values of <0.05 were considered indicate statistically significant differences.

### Reporting summary

Further information on research design is available in the Nature Portfolio Reporting Summary linked to this article.

## Data availability

The RNAseq data generated in this study are available in the GEO database under accession code GSE210182 and GSE210183. The publicly available TCGA data used in this study are available in the cbioportal database under following study; Metastatic Melanoma (UCLA, Cell 2016, https://www.cbioportal.org/study/summary?id= mel_ucla_2016), Melanoma (MSK, NEJM 2014, https://www. cbioportal.org/study/summary?id=skcm_mskcc_2014), Metastatic Melanoma (DFCI, Science 2015, https://www.cbioportal.org/study/ summary?id=skcm_dfci_2015), Colorectal Adenocarcinoma (TCGA, Firehose Legacy, https://www.cbioportal.org/study/summary?id= coadread_tcga), Colorectal Adenocarcinoma (TCGA, PanCancer Atlas, https://www.cbioportal.org/study/summary?id=coadread_ tcga_pan_can_atlas_2018), Ovarian Serous Cystadenocarcinoma (TCGA, Firehose Legacy, https://www.cbioportal.org/study/ summary?id=ov_tcga) and Ovarian Serous Cystadenocarcinoma (TCGA, Nature 2011, https://www.cbioportal.org/study/summary? id=ov_tcga_pub). The publicly available ChIPseq data used in this study are available in the ENCODE database under following accession code; (ENCFF076DCC, https://www.encodeproject.org/ experiments/ENCSR000ANQ/), (ENCFF219JIE, https://www. encodeproject.org/experiments/ENCSR086FIZ/), (ENCFF225RZA, https://www.encodeproject.org/experiments/ENCSR051VDI/), (ENCFF063WGZ, https://www.encodeproject.org/experiments/ ENCSR570YMM/) and (ENCFF042INO, https://www.encodeproject. org/experiments/ENCSR000CEL/). Source data are provided with this paper. The remaining data are available within the Article, Supplementary Information or Source Data file. Source data are provided with this paper.

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

## Acknowledgements

This work was supported by the National Research Foundation of Korea (NRF) grant funded by the Korean Government (MSIT) (No. NRF-2022R1A2C2009077 and 2022R1A5A2018865 to J.-H.C.; NRF-2020R1A5A8019180 and RS-2023-00224011 to H.-J.K.).

## Author contributions

M.P., Y.-S.K., J.-H.A., and R.H.P. designed and performed in vitro and in vivo experiments, analyzed data, and wrote the manuscript; Y.G., S.M., R.H.P., and Y.H. designed, performed, and analyzed cellular experi-ments; Y.-S.K and J.-W.N. performed metabolomics analyses; M.S.J., J.-H.A., and Y.H. carried out the TEM analysis; Y.-T.O. and B.K. contributed to the collection of human specimens; J.-O.K. and J.-W.N. reviewed and edited the manuscript; H.-J.K and J.-H.C. supervised the project designed the experiments, co-wrote the manuscript, and provided overall direction.

## Competing interests

The authors declare no competing interests.
