## [Peer Review File · Nature Communications]

Methionine consumption by cancer cells drives a progressive upregulation of PD-1 expression in CD4 T cellsEditorial Note:

Parts of this Peer Review File have been redacted as indicated to remove third-party material where no permission to publish could be obtained.

The figure on the last page in this Peer Review File is reproduced with permission from Elsevier

REVIEWER COMMENTS

Reviewer #1 (Remarks to the Author): with expertise in cancer immunology, metabolism

In this manuscript by Pandit et al., the authors propose a model whereby reduced methionine levels in the tumor microenvironment (TME) leads to a decrease in H3K79 methylation in CD4 T cells, including at the Prkaa1 promoter, resulting in reduced AMPK expression. In turn, this leads to ER stress and elevated expression of the spliced form of XBP1 (XBP1s), which induces PD-1 expression in CD4 T cells and dampens anti-tumor immunity. Methionine supplementation and/or pharmacologic activation of AMPK with AICAR reduced B16F10 tumor growth in C57BL/6J mice, but not RAG-KO mice, and was coupled with increased T cell infiltration into the tumor and reduced PD-1 expression in CD4 T cells. Moreover, genetic or pharmacologic inhibition of the IRE1-XBP1 axis reduced PD-1 expression in CD4 T cells in vitro. Finally, the authors propose that reduced methionine in the TME is due, in part, to consumption of methionine by cancer cells, and demonstrate that knockout of the methionine transporter, SLC43A2, in B16F10 cancer cells reduces tumor growth specifically in mice with mature T and B cells.

The manuscript is clearly written, and the main conclusions have important implications for tumor immunology and immunotherapy. Nonetheless, there are a few issues that need to be addressed to fully support the authors' conclusions before this manuscript can be considered for publication in Nature Communications.

Major comments:

1. The authors demonstrate that SLC43A2-KO B16F10 tumors grow slower in C57BL/6 mice and that CD4 T cells within the tumor exhibit reduced PD-1 protein expression. The explanation is that methionine uptake by cancer cells promotes CD4 T cell exhaustion; however, the authors did not measure methionine uptake or levels in their SLC43A2-KO cancer cells. The critical experiment here is to demonstrate the level of methionine uptake (i.e., ¹³C-methionine uptake) in the SLC43A2-KO tumor cells. If their hypothesis is correct, one would expect lower methionine uptake. If not, what is their explanation for the reduced PD-1 expression on T cells. If possible, methionine uptake in both tumor and T cells in a co-culture experiment would help resolve this issue.

2. Is the increase in XBP1 due to increased XBP1 splicing or an increase in total XBP1 levels? What is the ratio of spliced:unspliced XBP1 mRNA transcripts in CD4 T cells? The Novus Biologicals antibody used for immunoblotting should recognize both the spliced and unspliced form of XBP1, but the unspliced form was cropped out of the images. The authors should show both XBP1 bands.

3. In the co-culture experiment (Fig. 2d), it is unclear how the authors account for dilution of the CFSE dye due to cancer cell proliferation over their 36h time course? Typically in a mixed co-culture experiment different target cells are engineered to express different fluorescent proteins. In this case, how are CFSE-high versus CFSE-low cells differentiated as they are using the same fluorophore. In other words, is the ratio of B16F10 and TC-1 cells still 1:1 after 36 hr in culture in the control condition? Validation experiments documenting the level of CFSE dilution in B16 versus TC1 cells over the 36h timecourse must be provided to establish the validity of the assay.

4. While the authors cite a previous study that showed XBP1 binds the Pdcd1 promoter in CD8 T cells, direct evidence for this mechanism in CD4 T cells is lacking. Chromatin immunoprecipitation experiments for XBP1 binding to the Pdcd1 promoter would be required to make this statement in CD4 cells (see page 13). As a preamble to these ChIP studies, the authors can assess whether PD-1 mRNA expression is affected by the different metabolic, genetic, and pharmacologic perturbations in their study.

5. Does methionine supplementation or AICAR treatment affect PD-1 expression globally, or just in tumor-infiltrating CD4 T cells? The authors should include T cells isolated from the lymph node and/or spleen as a control for their tumor experiments to demonstrate specificity of their observations.

6. Are SLC43A2-KO tumors more sensitive to AMPK activation (i.e., AICAR treatment)? The inference in the paper is that methionine and AICAR synergize in tumor cells. In Figure 5g, the authors should use their KO tumors with methionine plus AICAR and track tumor growth.

7. Less CD4 and CD8 T cells are observed in tumors from mice with T cell-specific AMPK deletion (AMPK-KO) (Fig. 5e). Is this specific to the tumor or do AMPK-KO mice have less T cells systemically? For example, do T cell numbers in the lymph node differ between WT and AMPK-KO mice? Is PD-1 expression in lymph node-resident T cells different between WT and AMPK-KO mice?

8. If methylation affects the AMPK promoter to impact expression, what is the impact of DOTL1 inhibition affect AMPK mRNA expression?

9. The authors use the XBP1 inhibitor TUDCA for in vitro studies in Figure 6. Does treatment of tumor-bearing mice with TUDCA affect tumor growth and PD-1 expression on tumor-infiltrating CD4 T cells?

10. In the metabolomics data (Fig 4a), the authors show that CD4 T cells cultured in tumor-conditioned medium have increased glucose compared to control medium but decreased lactic acid. Can the authors speculate as to what these cells are doing with the increased glucose?

11. All tumor experiments are performed with a single cell line, B16F10. The authors should test a second cancer cell line to show that their observations are not specific to B16F10 cells.

Minor comments:

1. The figure order is hard to follow, as many are not in the same order in which they are referenced in the manuscript text.

2. Many immunoblots (e.g., Fig. 1, Fig 3, Fig. 4c, Extended Fig. 5c) appear to be overexposed. Can the authors please provide the original blots with lower exposures?

3. The authors include antibodies (e.g., anti-CD127, CD25, CD56, CD3) and other reagents (e.g., simvastatin) in their methods section that are not used in the present study. The methods should be updated accordingly. If these reagents were indeed used, it should be made clear where and how they were used.

Reviewer #2 (Remarks to the Author): with expertise in cancer immunology, metabolism

In this manuscript, Pandit et al showed an interesting mechanism of how metabolic changes in the TME impact anti-tumor response on CD4 T cells. They demonstrated that a shortage of methionine in the TME increased PD-1 expression in CD4 T cells. The expression of PD-1 is inhibited via methionine-triggered H3k79me2-AMPK axis activation. Overall, we found it is an intriguing study since not so many studies show how CD4 T cells are undergoing exhaustion via metabolic change and what is the impact of CD4 exhaustion on anti-tumor response. However, we doubt that the cell population they analyzed was not correct and could not decipher the real circumstance. Below, there are our major and minor comments:

Major:

1. In the main text line 90, line 233, and figure 1, the author wrote that they analyzed the PD-1 or AMPKa1 expression in immune cells from draining lymph nodes. It seems that they were using T cells

from draining lymph nodes in the whole study. However, cells from draining lymph nodes cannot be defined as in the tumor microenvironment and are also not the ideal population to study the impact of metabolic changes in the B16 tumor model. The authors should clarify whether it was the wrong texting. If they did use draining lymph nodes, they should not write tumor-infiltrating T cells and the whole concept written in this manuscript should be re-visited. The authors should analyze tumor-infiltrating lymphocytes (TILs) instead and provide data by making the comparison with draining lymph nodes.

2. In figure 2b, 2c and figure 5c, the tumor mass in each group was between 8 to 10 g. Some of the tumors in figure 2c and 5c were even over 10 g which was almost close to the half size of C57BL/6 mice. Generally, the tumor size was too huge and not as common as the normal size of the B16 tumor. We doubt that those are not the real size of the tumors otherwise the mice were severely suffered or dead. The author should carefully monitor the tumor growth and provide the images of the tumor with a precise scale.

3. In Figure 5, the author showed that AMPK KO on CD4 T cells impaired anti-tumor response by a decreased percentage of CD4 and CD8 and a reduction of CD8 cytotoxicity. The author further showed that methionine supplementation and combination with an AMPK activator (AICAR) drastically enhance antitumor immunity. This part is clear and intriguing. It raises the question that how AMPK or methionine affects CD4 helper cells anti-tumor response. Since Th1 is one major subset known for anti-tumor immunity, is it possible that AMPK influences T-bet expression and preferring for Th1 skewing? If it is the case, it is different what most literatures reported. How can the authors consolidate the findings and explain the discrepancies?

4. In this whole paper, the author only showed PD-1 expression as a CD4 exhaustion marker. Did the author also check other inhibitory markers like Tim-3 and Lag-3 expression?

Minor:

1. There are several typos in the figure legends. For example "thrice" in figure 1.

2. Can the authors also provide the absolute cell count data but not only cell percentage in figure 5?

Reviewer #3 (Remarks to the Author): with expertise in T cells, cancer immunology, ER stress

The manuscript by Pandit et al is an interesting and well performed study on mechanisms of exhaustion in CD4 T cells. The authors posit that little is known about the metabolic mechanisms regulating PD-1 expression in the TME. They demonstrate the role of methionine. They report that the upregulation of PD-1 expression in CD4, but not CD8 T cells, is associated with reduced levels of essential amino acids and that supplementation with methionine corrects the defect. They also show that methionine deficiency (and PD1 upregulation) is the result of nutrient competition between cancer cells and CD4 T cells. Therefore, the authors argue that methionine deficiency is responsible for immune incompetence in CD4 but not CD8 T cells in the TME. Using a step wise approach they go on to demonstrate the mechanism(s) underlining methionine deficit effects. They found that methionine negatively regulates H3K79me2 and this, in turn, impairs AMPK expression. Finally, they link these defects, and in particular PD1 upregulation, to excessive activation of XBP1, the downstream target of IRE1a, a conserved branch of the unfolded protein response in eukaryotes. In all, the study is meticulous and well-reasoned, and conclusion are by and large supported by the data. However, some aspects of the work raise concerns.

Major concerns

1. Fig. 4A. The heatmap shows that CD4 T cells treated with TM are enriched not only in fatty acids but also in glucose. This may seem counterintuitive, but since the general trust of this paper is that exhaustion is a deficit of methionine impinging upon AMPK and the inability of exhausted CD4 T cells to switch to the glycolytic pathway, it would have been important to underscore this finding. A consideration on the role of glycolysis versus fatty acid oxidation is missing in the paper but at the

very least it should be discussed particularly because a decrease in AMPK would also impinge (negatively) on fatty acid oxidation.

2. Fig 5E. These restoration experiments show that WT CD4 and CD8 T cells are equally positively affected. This seems to be contrary to the statement that methionine only affects CD4 T cells.

3. Fig 6C-D. This analysis is incomplete. It shows a band for XBP1s. However, the reagent used is a polyclonal antibody generated by immunization with "a genomic peptide made to an internal region of the human XBP1 protein (within residues 100-250). [Swiss-Prot P17861] This antibody is specific for both XBP1s and XBP1u" (from vendor website). Therefore, it is unclear what the band represents. I suggest that the experiments be repeated using a standard gel banding assay using specific primers showing both the unspliced and spliced forms of XBP1. This would not only make the results believable but also comparable to data from other groups.

4. Fig. 6F. The blot should indicate the molar concentration at which 4 μ 8c was used. This is not indicated in the figure legend or in the M&M. The authors should also show that in the experiment 4 μ 8c blocks/reduces XBP1 splicing not just PD1. As presented, this is inferred indirectly.

5. Fig. 6G. The blots show that in an unstimulated condition WT CD4 T cells have a very high basal level of spliced XBP1. This is surprising and brings into question the method used in the paper to detect XBP1s. As mentioned, this should be detected by gel banding and possibly by RT-qPCR.

6. Fig. 6I. While the use of TUDCA is per se appropriate, it comes as a surprise when the other panels of Fig. 6 use 4 μ 8c. If the authors want to use TUDCA, they should also show a direct effect on XBP1 splicing.

7. Fig 6I. The effect of genetic deletion of IRE1a/XBP1 on PD1 expression in activated CD4 T cells in the presence of TM shows a small effect by shXBP1 but not by shERN1 (IRE1a). If the effect on PD1 reflects activation of the IRE1a/XBP1 axis by TM, then both gene deletions should yield the same result on PD1 expression.

Minor points

8. In general figure legends lack definition of the different acronyms used in the figure themselves. This makes difficult to follow.

9. The dose of reagents used in vitro should be systematically specified.

10. Line 53-54. The sentence "on the importance of PD-1 blockade for increased PD-1 expression and the functional of CD4 T cells." Is missing something. Otherwise, it is not clear what the sentence intends to say.

Point-by-point responses to the reviewer's comments

REVIEWER 1

1. The authors demonstrate that SLC43A2-KO B16F10 tumors grow slower in C57BL/6 mice and that CD4 T cells within the tumor exhibit reduced PD-1 protein expression. The explanation is that methionine uptake by cancer cells promotes CD4 T cell exhaustion; however, the authors did not measure methionine uptake or levels in their SLC43A2-KO cancer cells. The critical experiment here is to demonstrate the level of methionine uptake (i.e., ¹³C-methionine uptake) in the SLC43A2-KO tumor cells. If their hypothesis is correct, one would expect lower methionine uptake. If not, what is their explanation for the reduced PD-1 expression on T cells. If possible, methionine uptake in both tumor and T cells in a co-culture experiment would help resolve this

Thank you for this suggestion. We conducted an uptake assay using methionine-d3 treatment (200 μ M) for 1 h. Intracellular methionine-d3 was detected using LC/MS (Line 523-529). The SLC43A2-KO B16F10 cells showed less methionine uptake than WT B16F10 cells (Supplementary figure 2j) (Line 143-146). Previous studies have shown that tumor cells outcompete T cells for methionine uptake to impair T cell function. SLC43A2 is a methionine transporter and SLC43A2-KO cells uptake a reduced amount of methionine from the media [1]. As SLC43A2-KO B16F10 uptakes less methionine, more methionine is available for tumor-infiltrating CD4 T cells. In line with this, in the *in vitro* study, we demonstrated reduced PD-1 expression in murine CD4 T cells (Fig 1f) and human CD4 T cells (Fig 1g) in tumor-conditioned medium after methionine supplementation. Also, Fig 2a showed reduced tumor growth and reduced PD-1 expression on tumor-infiltrating CD4 T cells subjected to methionine treatment in comparison to the vehicle (PBS-treated mice). Overall, we suggest that higher methionine availability for tumor-infiltrating CD4 T cells on SLC43A2-KO B16F10 tumor causes the reduced PD-1 expression in CD4 T cells.

<Supplementary Figure 2j>

2. Is the increase in XBP1 due to increased XBP1 splicing or an increase in total XBP1 levels? What is the ratio of spliced:unspliced XBP1 mRNA transcripts in CD4 T cells? The Novus Biologicals antibody used for immunoblotting should recognize both the spliced and unspliced form of XBP1, but the unspliced form was cropped out of the images. The authors should show both XBP1 bands.

Thank you for your comment. We have repeated western blotting experiments to quantify both spliced and unspliced XBP1 (Fig 6c, 6d, 6g, and 6i). XBP1 mainly increased due to an increase in total XBP1 levels, that is, both spliced XBP1 (XBP1s) and unspliced XBP1 (XBP1u) (Appendix Figure 1 a,b,c,d).

We replaced the blots with repeated results (Appendix Figure 1a to Fig 6c, 1b to 6d, 1c to 6g, and 1d to 6i) to show both the spliced and unspliced form of XBP1.

<Appendix Figure 1>

- Immunoblots showing XBP1s and XBP1u expression in CD4 T cells from tumor-free and tumor-bearing mice (left panel), the cultured CD4 T cells in CM and TM (middle panel), and the cultured CD4 T cells in CM and DM (right panel).
- Immunoblot of XBP1s and XBP1u in CD4 T cells cultured in CM, TM, and TM supplemented with methionine.
- Immunoblots showing XBP1s and XBP1u in activated or non-activated WT and AMPK KO CD4 T cells.
- Immunoblots for XBP1s and XBP1u expression in WT and AMPK KO CD4 T cells with or without an ER stress inhibitor (TUDCA).

3. *In the co-culture experiment (Fig. 2d), it is unclear how the authors account for dilution of the CFSE dye due to cancer cell proliferation over their 36h time course? Typically, in a mixed co-culture experiment different target cells are engineered to express different fluorescent proteins. In this case, how are CFSE-high versus CFSE-low cells differentiated as they are using the same fluorophore. In other words, is the ratio of B16F10 and TC-1 cells still 1:1 after 36 hr in culture in the control condition? Validation experiments documenting the level of CFSE dilution in B16 versus TC1 cells over the 36h time course must be provided to establish the validity of the assay.*

Thank you for highlighting this. You have raised an important point here. We have checked the method and corrected our description of 36 h of incubation to 6 h (Line 584-585). We apologize for our typing mistake.

To prepare the CTL target cells, TC-1 and B16F10 cells were labeled with high (5 μ M) and low (0.5 μ M) concentrations of CFSE. The CFSE high-labeled TC-1 cells were then mixed with CFSE low-labeled B16F10 cells in a ratio of 1:1, and 10^4 cells of the mixed target cells were cultured in vitro with or without CD8 T cells (5×10^6 cells) isolated from TC-1 tumor-bearing mice. CD4 T cells were also isolated from PBS- or methionine-treated TC-1 tumor-bearing mice and added to the respective groups. The cells were then cultured in vitro for 6 h at 37 °C before being analyzed using BD FACS Verse (BD Biosciences).

We used the CFSE dye in different concentrations (high and low) so that the peak for CFSE^{high}-labeled TC-1 (right) and CFSE^{low}-labeled B16F10 cells (left) in the histograms appeared at different intensities (Fig. 2d). A previous report also showed histogram peaks at different intensities for CFSE^{high}-labeled and CFSE^{low}-labeled cells [2]. Fig. 2d shows that the percentages of B16F10 and TC-1 cells in the control group were 50.9% and 49.1% and that the ratio of B16F10 to TC-1 cells was 1:1. We modified and stated additionally the above in Line 126-130.

4. *While the authors cite a previous study that showed XBPI binds the Pdc1 promoter in CD8 T cells, direct evidence for this mechanism in CD4 T cells is lacking. Chromatin immunoprecipitation experiments for XBPI binding to the Pdc1 promoter would be required to make this statement in CD4 cells (see page 13). As a preamble to these ChIP studies, the authors can assess whether PD-1 mRNA expression is affected by the different metabolic, genetic, and pharmacologic perturbations in their study.*

Thank you for your comment. We performed a ChIP assay using CD4 T cells and found that in TM samples, XBPIs binds to the PD-1 promoter (Supplementary Figure 8c). TM+Met samples

show reduced binding, suggesting that methionine prevents this binding (Supplementary Figure 8c). We stated this in Line 299-302.

<Supplementary Figure 8c>

5. Does methionine supplementation or AICAR treatment affect PD-1 expression globally, or just in tumor-infiltrating CD4 T cells? The authors should include T cells isolated from the lymph node and/or spleen as a control for their tumor experiments to demonstrate specificity of their observations.

Thank you for your comment. Methionine supplementation or AICAR treatment affects PD-1 expression only in tumor-infiltrating CD4 T cells (Fig. 5h) but not in CD4 T cells isolated from lymph nodes (Supplementary Figure 7i). We describe this in Line 257-260.

<Supplementary Figure 7i>

6. Are SLC43A2-KO tumors more sensitive to AMPK activation (i.e., AICAR treatment)? The inference in the paper is that methionine and AICAR synergize in tumor cells. In Figure 5g, the authors should use their KO tumors with methionine plus AICAR and track tumor growth.

Thank you for your comment. We transplanted WT mice with SLC43A2-KO B16F10 melanoma cells and treated the mice with PBS, methionine (40 mg/kg), and AICAR (500 mg/kg)

alone or in combination every alternate day. Combined treatment of methionine and AICAR more suppressed tumor growth than methionine or AICAR treatment alone (Supplementary Figure 7I), suggesting additive effect of methionine and AICAR for SLC43A2-KO tumor-bearing mice (Line 265 to 268).

<Supplementary Figure 7I>

Tumor picture (a), Tumor volume (b) and Tumor mass (c) of WT mice transplanted with SLC43A2-KO B16F10 melanoma cells followed by treatment with PBS or methionine (40 mg/kg) or AICAR (500 mg/kg) alone or in combination (n = 5 per group).

7. *Less CD4 and CD8 T cells are observed in tumors from mice with T cell-specific AMPK deletion (AMPK-KO) (Fig. 5e). Is this specific to the tumor or do AMPK-KO mice have less T cells systemically? For example, do T cell numbers in the lymph node differ between WT and AMPK-KO mice? Is PD-1 expression in lymph node-resident T cells different between WT and AMPK-KO mice?*

Our study of WT and T cell-specific AMPK-deleted mice revealed no significant differences in the frequencies of CD4 T cells and CD8 T cells in the spleen, peripheral lymph nodes, and mesenteric lymph nodes. This similarity in T cell phenotype between WT and AMPK-KO mice suggests that the absence of AMPK does not substantially affect T cell populations [3].

We have provided a comparison of CD4 and CD8 T cell populations between WT and AMPK KO mice from our previously published study (Supplementary Fig. 1c of reference [3]) as an appendix figure 2a. Also, PD-1 expression was similar in lymph node-resident T cells of WT and AMPK-KO mice (Appendix Figure 2).

< Supplementary Fig1c of reference [3] >

<Appendix Figure 2>

- CD4⁺ and CD8⁺ T cells frequencies and counts for indicated lymphoid tissues in WT and AMPK-KO mice
- PD-1 expression in secondary lymphoid organ resident CD4 T cells in WT and AMPK-KO mice

8. If methylation affects the AMPK promoter to impact expression, what is the impact of DOT1L inhibition affect AMPK mRNA expression?

You have raised an important point. In response to your question, we isolated CD4 T cells and cultured them in the presence or absence of a DOT1L inhibitor for 12 h. We then extracted RNA and performed RT-PCR, which revealed that DOT1L inhibition led to a significant reduction in AMPK mRNA expression (Supplementary figure 6e) (Line 241 to 243).

<Supplementary Figure 6e>

9. The authors use the XBP1 inhibitor TUDCA for *in vitro* studies in Figure 6. Does treatment of tumor-bearing mice with TUDCA affect tumor growth and PD-1 expression on tumor-infiltrating CD4 T cells?

Thank you for your comment. We transplanted B16F10 melanoma cells into WT mice and administered PBS or TUDCA every other day via intraperitoneal injection. Treatment with 150 mg/kg of TUDCA resulted in suppressed tumor growth (Supplementary figure 9a), decreased PD-1 expression (Supplementary Figure 9b), suggesting enhanced antitumor immunity (Line 305 to 309).

<Supplementary Figure 9a,b>

- WT mice transplanted with B16F10 melanoma cells and treated with TUDCA (150 mg/kg; intra-peritoneal) or control phosphate-buffered saline (PBS) every alternate day (n = 5 per group).
- PD-1 expression of tumor-infiltrating CD4.

10. In the metabolomics data (Fig 4a), the authors show that CD4 T cells cultured in tumor-conditioned medium have increased glucose compared to control medium but decreased lactic acid. Can the authors speculate as to what these cells are doing with the increased glucose?

Upon activation by TCR signaling, CD4 T cells undergo metabolic reprogramming in that they show increased glycolysis and downregulate fatty acid oxidation. However, our metabolite analysis showed that T cells cultured in TM showed fatty acids and glucose accumulation and a decrease in lactate, suggesting the inhibition of both glycolysis and fatty acid oxidation. These findings were supported by RNAseq data, which revealed a reduction in the expression of genes responsible for glycolysis and fatty acid oxidation in DM-cultured cells compared to those cultured in complete medium (CM) (Supplementary Figure 4f). The downregulation of AMPK

expression and activity in DM-treated CD4 T cells may have further reduced fatty acid oxidation and promoted fatty acid accumulation. Altogether, our results suggest that decreased glucose metabolism and fatty acid oxidation in CD4 T cells cultured in DM are associated with reduced AMPK expression resulting from methionine depletion. We added the result in the (Line 178) and dicussed in Dicussion section (Line 359-367).

<Supplementary figure 4f>

11. All tumor experiments are performed with a single cell line, B16F10. The authors should test a second cancer cell line to show that their observations are not specific to B16F10 cells.

Thank you for your suggestion. We assessed the effects of methionine treatment on TC-1 tumor-bearing mice and found that intratumoral injection of methionine resulted in suppressed tumor growth (Appendix Figure 3a) and reduced expression of PD-1 by tumor-infiltrating CD4 T cells (Appendix figure 3).

<Appendix Figure 3>

- Tumor volume (left) and Tumor weight (Right) of WT mice transplanted with TC-1 cells followed by treatment with PBS or methionine (40 mg/kg)
- PD-1 expression of tumor-infiltrating CD4.

We also assessed the effects of methionine treatment on WT mice that were transplanted with MC38 cancer cells followed by the treatment with PBS, methionine, AICAR, or a combination of methionine and AICAR every other day via intratumoral injection. We found that intratumoral injection of methionine suppressed tumor growth and reduced PD-1 expression on CD4 T cells. The combined treatment of methionine and AICAR further delayed tumor growth (Supplementary Figure 7e) and enhanced antitumor immunity by reducing PD-1 expression on CD4 T cells (Supplementary figure 7f). Hence, our observations are not specific to B16F10 cells (Line 257 to 259).

<Supplementary Figure 7g,h >

- The images of the tumor with a precise scale (left), tumor volume (middle) and tumor weight (right) of WT mice transplanted with MC38 cells followed by treatment with PBS or methionine (40 mg/kg) or AICAR (500 mg/kg) alone or in combination (n = 5 per group).
- PD-1 expression of tumor-infiltrating CD4 T cells.

Minor comments:

1. The figure order is hard to follow, as many are not in the same order in which they are referenced in the manuscript text.

Thank you for your comment. We have now arranged the figures in order as they are referenced in the manuscript.

2. Many immunoblots (e.g., Fig. 1, Fig 3, Fig. 4c, Extended Fig. 5c) appear to be overexposed. Can the authors please provide the original blots with lower exposures?

Thank you for your comment and suggestion. We have replaced the overexposed immunoblots with lower-exposed immunoblots in our revised manuscript.

3. *The authors include antibodies (e.g., anti-CD127, CD25, CD56, CD3) and other reagents (e.g., simvastatin) in their methods section that are not used in the present study. The methods should be updated accordingly. If these reagents were indeed used, it should be made clear where and how they were used.*

Thank you for your suggestion. We have made the relevant changes in the method section in our revised manuscript.

REVIEWER 2

1. *In the main text line 90, line 233, and figure 1, the author wrote that they analyzed the PD-1 or AMPK α 1 expression in immune cells from draining lymph nodes. It seems that they were using T cells from draining lymph nodes in the whole study. However, cells from draining lymph nodes cannot be defined as in the tumor microenvironment and are also not the ideal population to study the impact of metabolic changes in the B16 tumor model. The authors should clarify whether it was the wrong texting. If they did use draining lymph nodes, they should not write tumor-infiltrating T cells and the whole concept written in this manuscript should be re-visited. The authors should analyze tumor-infiltrating lymphocytes (TILs) instead and provide data by making the comparison with draining lymph nodes.*

When comparing tumor-bearing and tumor-free mice, we used draining lymph nodes as a reference for the tumor-free mice as they do not have tumor-infiltrating lymphocytes. However, for all other experiments where we compared mice with tumors, we used tumor-infiltrating lymphocytes.

2. *In figure 2b, 2c and figure 5c, the tumor mass in each group was between 8 to 10 g. Some of the tumors in figure 2c and 5c were even over 10 g which was almost close to the half size of C57BL/6 mice. Generally, the tumor size was too huge and not as common as the normal size of the B16 tumor. We doubt that those are not the real size of the tumors otherwise the mice were severely suffered or dead. The author should carefully monitor the tumor growth and provide the images of the tumor with a precise scale.*

Thank you for your critical comment. In the original experimental scheme approved by IACUC, we injected mice with 1×10^6 B16F10 cancer cells and started the treatment when the tumor size reached about 100–150 mm³ for 2 weeks. However, in the revised experiments (Fig. 2a,

2c, 2e, 2g, 5a, 5c, 5g, and 5k), we maintained a tumor volume of less than 2,000 mm³ at the time of mice sacrifice. For this, we subcutaneously injected mice with fewer cancer cells (3×10^5) and started the treatment when tumor volume reached around 50 mm³. Therefore, we could maintain a smaller tumor volume (<2000 mm³) and smaller tumor size (<5.5 g) (Appendix Figure 4a-h) at 22 days unless otherwise stated. We revised the Methods section for tumor inoculation and treatments (Line 531-540).

<Appendix Figure 4>

The images of the tumor with a precise scale (left), tumor volume (middle) and tumor weight (right) were shown.

- a. WT mice transplanted with B16F10 melanoma cells and treated with methionine (40 mg/kg; intratumoral injection) or control phosphate-buffered saline (PBS) every alternate day (n = 5 per group)
- b. Rag1^{-/-} mice transplanted with B16F10 melanoma cells and underwent PBS and methionine treatment (n = 3 per group).
- c. WT mice injected with SLC43A2-intact or SLC43A2 KO B16F10 cells (n = 5 per group).
- d. Rag1^{-/-} mice transplanted with SLC43A2-intact and SLC43A2 KO B16F10 cells (n = 3 per group).
- e. WT and AMPK KO mice (n = 5 per group).
- f. WT and AMPK KO mice transplanted with B16F10 melanoma cells and treated with methionine (40 mg/kg; intra-tumor) or phosphate-buffered saline (PBS) as a control every alternate day (n = 4 per group).
- g. WT mice transplanted with B16F10 melanoma cells followed by treatment with PBS or methionine (40 mg/kg) or AICAR (500 mg/kg) alone or in combination (n = 5 per group).
- h. Rag1^{-/-} mice transplanted with B16F10 melanoma cells followed by treatment with PBS or methionine (40 mg/kg) or AICAR (500 mg/kg) alone or in combination (n = 4 per group).

3. In Figure 5, the author showed that AMPK KO on CD4 T cells impaired anti-tumor response by a decreased percentage of CD4 and CD8 and a reduction of CD8 cytotoxicity. The author further showed that methionine supplementation and combination with an AMPK

activator (AICAR) drastically enhance antitumor immunity. This part is clear and intriguing. It raises the question that how AMPK or methionine affects CD4 helper cells anti-tumor response. Since Th1 is one major subset known for anti-tumor immunity, is it possible that AMPK influences T-bet expression and preferring for Th1 skewing? If it is the case, it is different what most literatures reported. How can the authors consolidate the findings and explain the discrepancies?

We appreciate your critical review. In our study, we discovered that methionine and AMPK regulate PD-1 expression in CD4 T cells, which contributes to anti-tumor immunity. Our previous study revealed that when the AMPK gene was deleted from CD4 T cells, there were no significant differences in CD4, CD8 (Supplementary Fig 1c of reference [3]), and Th1 cell (Supplementary Fig 1f of reference [3]) populations, as well as, in in vitro Th1 differentiation (Supplementary Fig 1h of reference [3]). Thus, we can conclude that the anti-tumor immunity conferred by AMPK in this study is not due to the expression of T-bet or Th1 cells. Supplementary Figure 1c, 1f and 1h of reference [3] provides a comparison of CD4 and CD8 T cell populations, Th1 cells and invitro Th1 differentiation between WT and AMPK KO mice.

As the anti-tumor response of CD4 T cells is not due to Th1 differentiation, the recovered activity of CD4 T cells from exhaustion and ER stress in association with decreased PD-1 expression on tumor-infiltrating CD4 T cells by methionine and AMPK may be responsible for the enhanced antitumor activity. Furthermore, previous studies by other researchers suggest that CD4 helper cells can enhance CD8 T cell proliferation and activity, prevent CD8 T cell exhaustion, and promote anti-tumor immunity [4-6].

<Supplementary Fig 1c, 1f, and 1h of reference [3]>

- The frequencies and counts of CD4 and CD8 T cells for indicated lymphoid tissues in WT and AMPK KO mice.
- IFN-γ⁺ (Th1) and IL-17A⁺ (Th17) CD4 T cells in WT and AMPK KO mice. In vitro Th1 differentiation of WT CD4 and AMPK KO CD4 in presence of IL-12 cytokine.

4. In this whole paper, the author only showed PD-1 expression as a CD4 exhaustion marker. Did the author also check other inhibitory markers like Tim-3 and Lag-3 expression?

Following your suggestion, we investigated the presence of other inhibitory markers, such as Tim-3 and Lag-3, in tumor-infiltrating CD4 T cells and CD4 T cells isolated from the lymph nodes of tumor-bearing mice. Our findings indicate that there are no significant differences in the expression of Tim-3 and Lag-3 in CD4 derived from lymph nodes of tumor bearing mice (Supplementary fig. 7h), as well as in tumor-infiltrating CD4 T cells (Supplementary Fig. 7i) of tumor-bearing mice after treatment with methionine or AICAR supplementation or methionine and AICAR cotreatment (Line 260 to 265).

- a. LAG-3 (left) and TIM-3 (right) expression in CD4 T cells derived from lymph nodes of tumor-bearing mice.
- b. LAG-3 (left) and TIM-3 (right) expression of tumor-infiltrating CD4 T cells.

Minor comments:

1. There are several typos in the figure legends. For example, “thrice” in figure 1.

Thank you. We found your comments extremely helpful and have revised them accordingly.

2. Can the authors also provide the absolute cell count data but not only cell percentage in figure 5?

Thank you for highlighting this. We have provided the absolute cell count for CD4 and CD8 T cells with supplementary Fig. 7e and 7f.

<Supplementary Figure 7e,f>

- a. Absolute cell count of tumor-infiltrating CD4 (left) and CD8 (right) T cells of WT and AMPK KO mice transplanted with B16F10 melanoma cells following treatment with methionine (40 mg/kg; intra-tumor) or phosphate-buffered saline (PBS) as a control every alternate days (n = 5 per group).
- b. Absolute cell count of tumor-infiltrating CD4 (left) and CD8 (right) T cells from mice transplanted with B16F10 cells melanoma cells followed by treatment with PBS or methionine (40 mg/kg) or AICAR (500 mg/kg) alone or in combination (n = 5 per group).

REVIEWER 3

1. Fig. 4A. The heatmap shows that CD4 T cells treated with TM are enriched not only in fatty acids but also in glucose. This may seem counterintuitive, but since the general trust of this paper is that exhaustion is a deficit of methionine impinging upon AMPK and the inability of exhausted CD4 T cells to switch to the glycolytic pathway, it would have been important to underscore this finding. A consideration on the role of glycolysis versus fatty acid oxidation is missing in the paper but at the very least it should be discussed particularly because a decrease in AMPK would also impinge (negatively) on fatty acid oxidation.

CD4 T cells activated by TCR signaling undergo metabolic reprogramming that favors glycolysis-dependent energy production and downregulates fatty acid oxidation. Through the metabolite analysis, we observed the accumulation of fatty acid and glucose, as well as, a decrease in lactate in T cells cultured in DM. These results suggest the inhibition of both glycolysis and fatty acid oxidation. The heatmap revealed a reduction in most of the genes responsible for glycolysis and fatty acid oxidation in cells cultured in DM than those cultured in CM. These results may account for the accumulation of fatty acid and glucose and lowered lactate in DM-cultured cells (Supplementary Figure 4f). Similar to TM, treatment with DM in CD4 T cells can cause downregulation of AMPK, lowering its activity, decreasing fatty acid oxidation, and promoting the accumulation of fatty acid [7]. Overall, we suggest that the decreased glucose metabolism and fatty acid oxidation in CD4 T cells cultured in DM may be associated with the reduction of AMPK expression resulting from methionine depletion. We added the result in the (Line 178) and discussed in Discussion section (Line 359 to 367).

<Supplementary Figure 4f>

2. Fig 5E. These restoration experiments show that WT CD4 and CD8 T cells are equally positively affected. This seems to be contrary to the statement that methionine only affects CD4 T cells.

Thank you for highlighting this. Our study does not conclude that methionine affects CD4 T cells alone. Rather, it concludes that methionine affects PD-1 expression in CD4 T but not CD8 T cells. Moreover, in our tumor inoculation experiment on WT mice, both CD4 and CD8 T cell populations were affected. Therefore, we established that the anti-tumor response of CD4 T cells is due to reduced CD4 T cell exhaustion resulting from decreased PD-1 expression on tumor-infiltrating CD4 T cells and the reduction of ER stress by methionine and AMPK. In addition, CD4 helper T cells affect CD8 proliferation and activity, as well as prevent CD8 T cell exhaustion and increase anti-tumor immunity [4-6]. This may be why CD8 T cells were also affected in our study.

3. Fig 6C-D. This analysis is incomplete. It shows a band for XBP1s. However, the reagent used is a polyclonal antibody generated by immunization with “a genomic peptide made to an internal region of the human XBP1 protein (within residues 100-250). [Swiss-Prot P17861] This antibody is specific for both XBP1s and XBP1u” (from vendor website). Therefore, it is unclear what the band represents. I suggest that the experiments be repeated using a standard gel banding assay using specific primers showing both the unspliced and spliced forms of XBP1. This would not only make the results believable but also comparable to data from other groups.

Thank you for your feedback. We have repeated western blotting experiments to show both spliced and unspliced XBP1 (Fig 6c, 6d, 6g, and 6i). XBP1 mainly increased due to an increase in total XBP1 levels, that is, both spliced XBP1 and unspliced XBP1 (Appendix Figure 5a-d). We replaced the blots with repeated results (Appendix Figure 6a to Fig 6c, 6b to 6d, 6c to 6g, and 6d to 6i) to show both the spliced and unspliced form of XBP1.

<Appendix Figure 5>

- e. Immunoblots showing XBP1s and XBP1u expression in CD4 T cells from tumor-free and tumor-bearing mice (left panel), the cultured CD4 T cells in CM and TM (middle panel), and the cultured CD4 T cells in CM and DM (right panel).
- f. Immunoblot of XBP1s and XBP1u in CD4 T cells cultured in CM, TM, and TM supplemented with methionine.
- g. Immunoblots showing XBP1s and XBP1u in activated or non-activated WT and AMPK KO CD4 T cells.
- h. Immunoblots for XBP1s and XBP1u expression in WT and AMPK KO CD4 T cells with or without an ER stress inhibitor (TUDCA).

4. Fig. 6F. The blot should indicate the molar concentration at which 4μ8c was used. This is not indicated in the figure legend or in the M&M. The authors should also show that in the experiment 4μ8c blocks/reduces XBP1 splicing not just PD1. As presented, this is inferred indirectly.

Thank you for your feedback. The molar concentration of 4μ8c used was 10 μM; we have included this information in the revised manuscript (Line 903). As per your suggestion, we have performed western blotting to check XBP1 splicing and found that 4μ8c prevented XBP1 splicing (Appendix Figure 6).

<Appendix Figure 6>

5. Fig. 6G. The blots show that in an unstimulated condition WT CD4 T cells have a very high basal level of spliced XBP1. This is surprising and brings into question the method used in the paper to detect XBP1s. As mentioned, this should be detected by gel banding and possibly by RT-qPCR.

In the main manuscript Figure 6g, the blots showed that compared with stimulated cells, WT CD4 T cells had a very high basal level of spliced XBP1 when unstimulated. This may be due to increased protein loading, as the expression of β-actin is also higher under unstimulated conditions. We also repeated this experiment (Appendix Figure 7) and replaced the figure in our revised manuscript (Figure 6g).

<Appendix Figure 7>

6. Fig. 6I. While the use of TUDCA is per se appropriate, it comes as a surprise when the other panels of Fig. 6 use 4μ8c. If the authors want to use TUDCA, they should also show a direct effect on XBP1 splicing.

We observed an increase in ER stress in AMPK KO CD4 T cells, as evidenced by higher levels of both PD-1 and XBP1 compared to WT CD4 T cells. To investigate the direct effect of the well-known ER-stress inhibitor TUDCA on XBP1 splicing in both WT and AMPK-KO CD4 T cells, we repeated the experiment (Appendix Figure 8). TUDCA is known to increase PD-1 and XBP1 splicing. Previous studies have reported a direct correlation between higher XBP1 levels and higher PD-1 expression [8]. To investigate the relationship between PD-1 expression and XBP1 splicing in our study, we used 4μ8c, a specific IRE1α inhibitor. IRE1α occurs upstream of XBP1, and previous studies have reported that functional XBP1s is generated by IRE1α [9, 10]. To specifically target XBP1, we used 4μ8c instead of TUDCA in the subsequent experiments. We replaced blots of XBP1 in new Fig. 6i with this result (Line 305-307).

<Appendix Figure 8>

7. Fig 6L. The effect of genetic deletion of IRE1a/XBP1 on PD1 expression in activated CD4 T cells in the presence of TM shows a small effect by shXBP1 but not by shERN1 (IRE1a). If the effect on PD1 reflects activation of the IRE1a/XBP1 axis by TM, then both gene deletions should yield the same result on PD1 expression.

Thank you for your feedback. We repeated the experiment and conducted flow cytometry to clarify the reduction of PD-1 expression in shXBP1 and shERN1. We found an increase in PD-1 expression after culture in TM (Appendix Figure 9), and a reduction of PD-1 expression was found by shXBP1 and shERN1 (IRE1a) infection (Appendix Figure 9). The reduction in PD-1 level was similar in the shXBP1 and shERN1 KD groups (Appendix Figure 9). The significance of each comparison is shown below. We additionally noted the p-value on Fig 6l obtained from one-way ANOVA comparison of PD-1 MFI between shXBP1- and shIRE1a-infected CD4 T cells (Line 315-316).

<Appendix Figure 9>

Minor points

8. *In general figure legends lack definition of the different acronyms used in the figure themselves. This makes difficult to follow.*

Thank you for your feedback. We have revised the figure legends accordingly and added the definition of the acronyms in the revised manuscript.

9. *The dose of reagents used in vitro should be systematically specified.*

Thank you for your feedback. We have specified the doses of every reagent used in the experiments in the materials and methods section in the revised manuscript.

10. *Line 53-54. The sentence “on the importance of PD-1 blockade for increased PD-1 expression and the functional of CD4 T cells.” Is missing something. Otherwise, it is not clear what the sentence intends to say.*

Thank you so much for highlighting this. We have revised the sentence as:

Although CD8 T cell exhaustion has been well-established and considered a target of immunotherapy for cancer patients, relatively few studies have focused on the importance of PD-1 blockade for increased PD-1 expression and the functional recovery of CD4 T cells (Line 52 to 53).

Reference

1. Bian, Y., et al., *Cancer SLC43A2 alters T cell methionine metabolism and histone methylation*. Nature, 2020. **585**(7824): p. 277-282.
2. Bustos-Morán, E., et al., *Aurora A controls CD8+ T cell cytotoxic activity and antiviral response*. Scientific Reports, 2019. **9**(1): p. 2211.
3. Pandit, M., et al., *AMPK suppresses Th2 cell responses by repressing mTORC2*. Experimental & Molecular Medicine, 2022. **54**(8): p. 1214-1224.
4. Lai, Y.-P., et al., *CD4+ T cell-derived IL-2 signals during early priming advances primary CD8+ T cell responses*. PloS one, 2009. **4**(11): p. e7766.
5. Kitchen, S.G., et al., *CD4 on CD8+ T cells directly enhances effector function and is a target for HIV infection*. Proceedings of the National Academy of Sciences, 2004. **101**(23): p. 8727-8732.
6. Lu, Y.-J., et al., *CD4 T cell help prevents CD8 T cell exhaustion and promotes control of Mycobacterium tuberculosis infection*. Cell reports, 2021. **36**(11): p. 109696.
7. Herzig, S. and R.J. Shaw, *AMPK: guardian of metabolism and mitochondrial homeostasis*. Nature reviews Molecular cell biology, 2018. **19**(2): p. 121-135.
8. Ma, X., et al., *Cholesterol induces CD8+ T cell exhaustion in the tumor microenvironment*. Cell metabolism, 2019. **30**(1): p. 143-156. e5.
9. Wu, R., et al., *Involvement of the IRE1 α -XBP1 pathway and XBP1s-dependent transcriptional reprogramming in metabolic diseases*. DNA and cell biology, 2015. **34**(1): p. 6-18.
10. Yoshida, H., et al., *XBP1 mRNA is induced by ATF6 and spliced by IRE1 in response to ER stress to produce a highly active transcription factor*. Cell, 2001. **107**(7): p. 881-891.

REVIEWERS' COMMENTS

Reviewer #1 (Remarks to the Author):

The authors have sufficiently addressed my concerns with their revision.

Reviewer #2 (Remarks to the Author):

The authors address my original concerns. The revised manuscript indeed improves the clarity and results support the conclusion.

Reviewer #3 (Remarks to the Author):

The authors have addressed the comments. However, there remains a question concerning the Western blot data shown in Fig. 6.

Repeatedly, XBP1u (70kD) is shown as a band of lower MW relative to XBP1s (50kD). I think that this requires explanation or revision of Fig. 6.

Editorial Note:

The first figure below is redacted as no permission to publish could be obtained.

Figure below reproduced from Ma, X., Bi, E., Lu, Y. et al. Cholesterol Induces CD8+ T Cell Exhaustion in the Tumor Microenvironment. *Cell Metabolism*. **30**, 143-156.e5 (2019), with permission from Elsevier.

Reviewer #3 (Remarks to the Author):

The authors have addressed the comments. However, there remains a question concerning the Western blot data shown in Fig. 6. Repeatedly, XBP1u (70kD) is shown as a band of lower MW relative to XBP1s (50kD). I think that this requires explanation or revision of Fig. 6.

Response to reviewer's question:

Thank you for your concern. After transcription, XBP1 undergoes splicing to form two isoforms: XBP1u (unspliced form) and XBP1s (spliced form). XBP1u is composed of 267 amino acids, while XBP1s is composed of 371 amino acids. During unconventional splicing by IRE1a, 26 nucleotides are removed in the intron region of unspliced mRNA, resulting in a translational frameshift that produces the XBP1s protein. In Western blot analysis, XBP1u is smaller in size than XBP1s and therefore migrates closer to the bottom of the SDS-PAGE gel. XBP1s is larger in molecular weight than XBP1u due to the addition of 104 amino acids to the C-terminus. Please refer to the following data sheet, which is offered by the manufacturer (Novus Biosciences, NBP1-77681), and the result of XBP1 western blot from the reference (*Cell Metabolism*. 2019 Jul 2;30(1):143-156.e5.).

[redacted]